# Cortico-subcortical β burst dynamics underlying movement cancellation in humans

**Darcy A Diesburg[1]\*, Jeremy DW Greenlee[2,3], Jan R Wessel[1,3,4]\***

[1]Department of Psychological and Brain Sciences, University of Iowa, Iowa City, United States; [2]Department of Neurosurgery, University of Iowa Carver College of Medicine, Iowa City, United States; [3]Iowa Neuroscience Institute, University of Iowa, Iowa City, United States; [4]Department of Neurology, University of Iowa Carver College of Medicine, Iowa City, United States

**Abstract** Dominant neuroanatomical models hold that humans regulate their movements via loop-like cortico-subcortical networks, which include the subthalamic nucleus (STN), motor thalamus, and sensorimotor cortex (SMC). Inhibitory commands across these networks are purportedly sent via transient, burst-like signals in the β frequency (15–29 Hz). However, since human depth-recording studies are typically limited to one recording site, direct evidence for this proposition is hitherto lacking. Here, we present simultaneous multi-site recordings from SMC and either STN or motor thalamus in humans performing the stop-signal task. In line with their purported function as inhibitory signals, subcortical β-bursts were increased on successful stop-trials. STN bursts in particular were followed within 50 ms by increased β-bursting over SMC. Moreover, between-site comparisons (including in a patient with simultaneous recordings from SMC, thalamus, and STN) confirmed that β-bursts in STN temporally precede thalamic β-bursts. This highly unique set of recordings provides empirical evidence for the role of β-bursts in conveying inhibitory commands along long-proposed cortico-subcortical networks underlying movement regulation in humans.

**\*For correspondence:**
darcy-diesburg@uiowa.edu (DAD);
jan-wessel@uiowa.edu (JRW)

**Competing interest:** The authors declare that no competing interests exist.

## Editor's evaluation

This work makes an important contribution to the literature and addresses timely and interesting questions relating to the role of transient beta oscillations in cancelling motor responses in a rare and valuable dataset.

## Introduction

Movement cancellation – that is, the ability to stop ongoing or prepotent movements when necessary – allows humans to adapt their behavior quickly to changing environmental demands. A predominant paradigm used to investigate inhibitory control is the stop-signal task (SST), wherein participants are tasked with executing and sometimes cancelling movements (*Logan et al., 1984*; *Verbruggen et al., 2019*). This task allows for computation of the duration of the latent cancellation process (stop-signal reaction time, SSRT), although no overt response is made when participants successfully stop (*Verbruggen,, 2008*; *Boucher et al., 2007*.) The neural pathways underlying movement cancellation comprise a fronto-basal ganglia (FBg) network for inhibitory control (*Wessel and Aron, 2017*), which recruits known anti-kinetic basal ganglia pathways (*Jahanshahi et al., 2015*). When a stop-signal occurs, the right inferior frontal cortex (rIFC) purportedly excites the subthalamic nucleus (STN) via a monosynaptic 'hyperdirect' pathway between the two regions (*Nambu et al., 2002*; *Aron, 2007*;

*Chen et al., 2020*). Subsequently, the STN broadly excites the internal segment of the globus pallidus (GPi; *Parent and Hazrati, 1993*; *Gillies and Willshaw, 1998*), the output nucleus of the basal ganglia. In turn, the GPi inhibits the ventral oral posterior (Vop) region of the motor thalamus (*Inase and Tanji, 1995*; *Kuo and Carpenter, 1973*). It has been proposed that the resultant net-inhibition of thalamocortical signaling loops (i.e., motoric loops between thalamus and sensorimotor cortex) enables the type of rapid movement cancellation found in tasks like the SST (*Parent and Hazrati, 1993*; *Jahanshahi et al., 2015*).

Recordings from nodes of this FBg network have revealed that communication through these pathways likely occurs in the β frequency band. During movement execution, decreases in averaged β power are observed over sensorimotor cortex (SMC; both intracranially and on the scalp, *Crone et al., 1998*; *Pfurtscheller and Lopes da Silva, 1999*; *Kühn et al., 2004*; *Takemi et al., 2013*) and in subcortical motor regions such as the STN  (*Alegre et al., 2005*) and the ventral intermediate (VIM) nucleus of the motor thalamus, a part of motor thalamus adjacent to Vop (*Basha et al., 2014*). In contrast, averaged β power is *increased* in SMC and STN when inhibitory control is required, both following stop-signals in the SST (*Wessel et al., 2016* ; *Swann et al., 2009*; *Swann et al., 2011*; *Ray et al., 2012*; *Alegre et al., 2013*; *Benis et al., 2014*; *Bastin et al., 2014*) and during motor conflict more broadly (*Brittain et al., 2012*; *Wessel et al., 2019*). Similar increases in β power during movement cancellation are observed in cortical regions ostensibly upstream of the STN and thalamus, such as the pre-supplementary motor area (*Swann et al., 2012*; *Picazio et al., 2014*) and the rIFC (*Swann et al., 2009*). Together, these findings have established cortical and subthalamic β activity as an index of inhibitory control.

However, cross-species research has revealed that these changes in β power do not reflect *sustained* β oscillations at the single-trial level (*Feingold et al., 2015*; *Sherman et al., 2016*; *Shin et al., 2017*; *Tinkhauser et al., 2017*; *Maling et al., 2018*; *Cagnan et al., 2019*). Unaveraged β activity has properties better characterized as intermittent bursting instead of slow-and-steady modulations of amplitude (*van Ede et al., 2018*). In line with this, β bursts are more predictive of behavior than fluctuations in averaged β power. For example, perceptual stimuli preceded closely by β bursts in somatosensory cortex are less likely to be detected (*Shin et al., 2017*) and β bursts in motor cortex closely preceding imperative stimuli are associated with slower responses (*Little et al., 2019*). Biophysical computational models suggest that these bursts in SMC relate to coincident proximal and distal excitatory drives to the synapses of neocortical pyramidal neurons (*Sherman et al., 2016*). Thus, not only do β bursts carry fine-grained information about behavior on the single-trial level, they also relate more closely to underlying mechanisms than averaged β. Notably, two recent studies have demonstrated that β bursts on the scalp relate to the inhibitory aspects of movement regulation. One study demonstrated reductions in β burst rates over SMC during go trials, as well as increases in burst rates over frontocentral and motor cortices during stop trials, and found that successful stop trials featured a greater number of frontocentral β bursts before SSRT on average than failed stop trials (*Wessel, 2020*). A subsequent study by *Jana et al., 2020* demonstrated that β bursts over prefrontal cortex were followed within 20ms by broad skeleto-motor suppression and within 40ms by outright cancellation detectable at the motor effector.

While these studies identify potential (pre)frontal cortical control signals associated with movement cancellation, they are uninformative regarding the downstream basal ganglia-thalamic dynamics through which inhibitory control of SMC is ostensibly implemented. Although transient β bursts are known to exist in the STN (*Torrecillos et al., 2018*; *Lofredi et al., 2019*), it is unclear what functional role subcortical β bursts play during movement regulation, and whether their dynamics conform to the dominant neurophysiological and neuroanatomical models of inhibitory control. Beyond generating basic knowledge about the neurophysiology of basal ganglia motor circuitry, elucidating these dynamics would also greatly inform therapeutic approaches that are already targeting the known pathological β bursting that occurs in these subcortical regions (*Tinkhauser et al., 2017*; *Little and Brown, 2020*).

Our aims for the current study were twofold. Firstly, we investigated whether β bursts in subcortical regions of basal ganglia-thalamic inhibitory pathways are associated with movement cancellation. To this end, we investigated the relationship between SST performance and β burst rates in both STN and motor thalamus. Furthermore, we tested whether these subcortical bursts have reliable temporal relationships with movement-related β bursts in SMC, suggestive of an inhibitory influence of the

**Table 1.** Means of stop-signal task behavioral performance metrics.
* Indicates significant difference between thalamic and STN groups at p < 0.0001. SD = standard deviation.

| Reaction times (ms) | | | Accuracy | |
|---|---|---|---|---|
| Go | Failed stop | SSRT | Go trials* | Stop trials |
| All participants | | | | |
| 943 (SD: 213) | 763 (SD: 174) | 474 (SD: 232) | 0.89 (SD: 0.07) | 0.61 (SD: 0.16) |
| STN DBS | | | | |
| 952 (SD: 234) | 786 (SD: 213) | 533 (SD: 296) | 0.83 (SD: 0.05) | 0.55 (SD: 0.18) |
| Thalamic DBS | | | | |
| 934 (SD: 203) | 741 (SD: 136) | 421 (SD: 150) | 0.94 (SD: 0.04) | 0.65 (SD: 0.11) |

subcortical regions on SMC. Secondly, we evaluated existing models of inhibitory control networks by assessing relative timing of bursts across subcortical recording sites. The dominant model of a fronto-basal ganglia circuit for inhibitory control suggests that movement cancellation is accomplished by net-inhibition of the motor thalamus by STN (*Jahanshahi et al., 2015*). Hence, in line with the proposition that STN is recruited before the thalamus during cancellation, we expected β bursts related to movement cancellation to emerge first in the STN, followed by bursts in the thalamus.

We collected recordings of local field potentials (LFPs) simultaneously from SMC and a subcortical site (either the STN or motor thalamus) during awake deep-brain stimulation (DBS) lead implantation surgery in two groups of patients: patients with Parkinson's disease (PD) undergoing STN implantation and essential tremor (ET) patients undergoing implantations in the motor thalamus. Moreover, data from one highly unique PD patient included simultaneous recordings from all three locations: SMC, STN, and motor thalamus. During the recordings, patients performed an auditory version of the SST to test their ability to rapidly cancel movements. This study leverages unique multi-site intracranial recordings and a cognitive paradigm to investigate the circuit dynamics of cortical and subcortical β bursts during movement cancellation.

## Results
### Behavior
While LFPs were recorded from SMC and either STN or thalamus, 21 participants completed an auditory SST (see *Figure 3—figure supplement 1*). Behavioral results are shown in *Table 1*. To confirm that patients' diagnoses did not affect cognitive performance or task strategies in a way that would preclude comparing the two participant groups, we compared behavioral performance between the STN and thalamic groups. Go accuracy was the only behavioral metric which significantly differed between the STN and thalamic groups, with thalamic patients responding more accurately on go-trials (94% vs 83% accuracy, $T(19) = 5.22$, p < 0.0001, $d = 2.27$). Numerically, the thalamic implant group also responded faster during correct go and failed stop trials and cancelled movements more quickly than STN DBS patients, as indicated by SSRT (which was in the typical elongated range for movement disorder patients: *Gauggel et al., 2004*; *Obeso et al., 2011*; *Hughes et al., 2019*). However, none of these results were significant, indicating comparable task performance between both groups. (Go RT: $T(19) = -0.19$, p = 0.85, $d = 0.08$; Failed stop RT: $T(19) = -0.58$, p = 0.57, $d = 0.25$; SSRT: $T(19) = -1.11$, $P = 0.28$, $d = 0.48$; Stop accuracy: $T(19) = 1.59$, $P = 0.13$, $d = 0.68$.)

### Averaged event-related spectral perturbation (ERSP) analysis
To confirm accurate electrode placement over hand-related areas of SMC, event-related spectral perturbation (ERSP) was quantified from –100 preceding to 1500 ms following the go-signals in a go-only localizer task. This short, 40-trial block was identical to the main stop-signal task but *did not contain stop signals* (in other words, it was a speeded two-choice response task) and was administered before the subcortical lead placement. While in the OR, we then visually checked the ERSPs for a decrease in average β band amplitude, a known signature of movement-related activity in SMC (cf.,

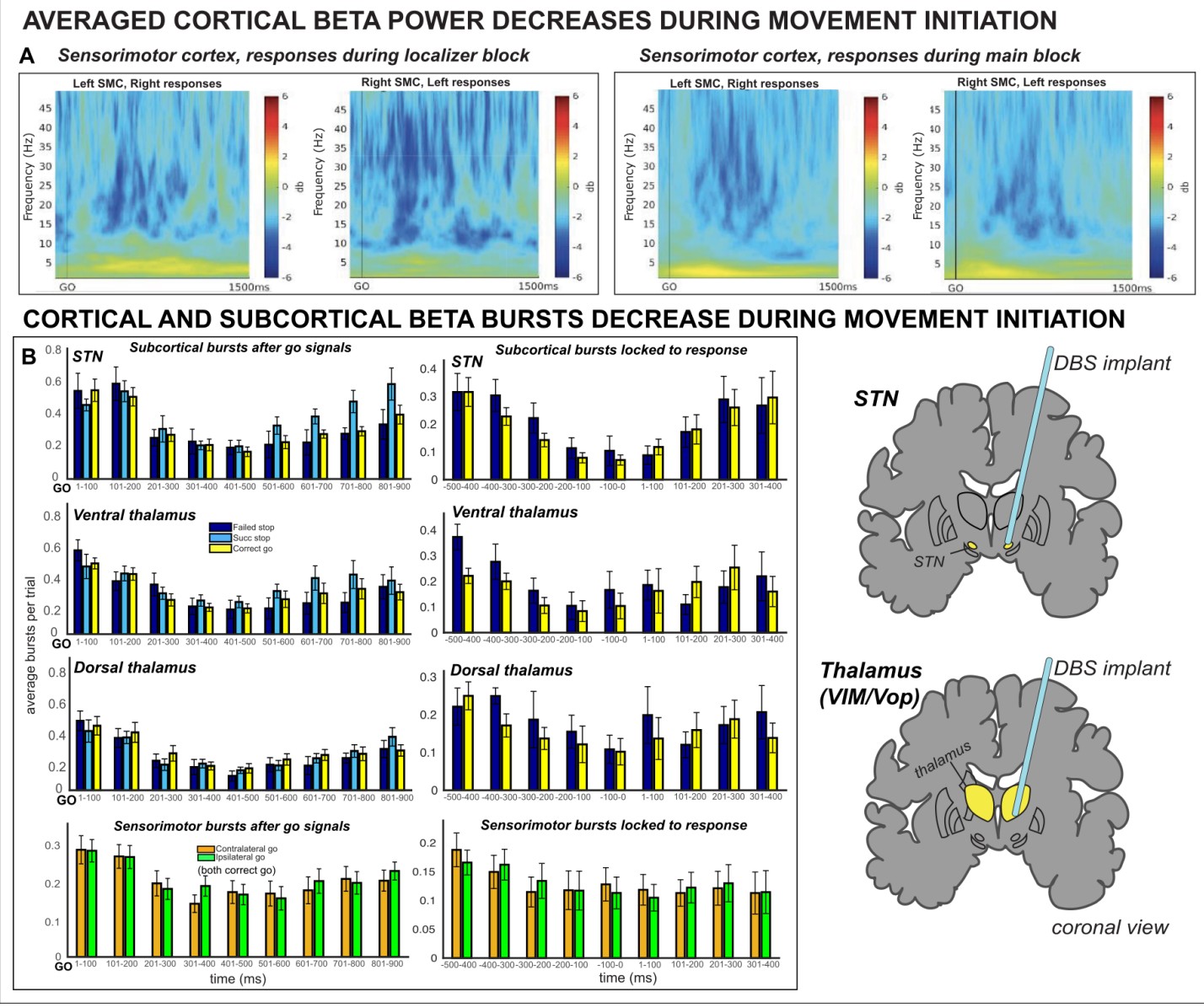

**Figure 1.** Averaged β power and β burst rates decreased across recording sites during movement execution. (**A**) Broadband ERSPs shown were computed from 100ms preceding to 1500ms following the go signal. The two left-most plots include ERSPs from the localizer block and the right-most plots include ERSPs during the main block. In averaged ERSPs during movement on go-trials, both localizer and main task sessions show clearly visible β power decreases following the go signal. (**B**) Average burst rates at each recording location following the go-signal and surrounding response execution (for go and failed stop trials) are depicted in time bins of 100ms. During the main task session, β burst rates decrease quickly following the go signal in STN, thalamus (both ventral and dorsal contacts), and SMC until a response is made.

The online version of this article includes the following figure supplement(s) for figure 1:

**Figure supplement 1.** Averaged ERSPs from all SMC contact pairs, 1000 ms before to 1000 ms following movement execution during contralateral correct go trials.

*Crone et al., 1998*; *Pfurtscheller and Lopes da Silva, 1999*; *Kühn et al., 2004*; *Takemi et al., 2013*), to confirm SMC electrode placement for the main experiment. More information about the localization process can be found in the Methods section. We then quantitatively investigated these relationships after surgery and found decreases in averaged β band (15–29 Hz) power observed at contralateral (to the response) SMC sites following go signals (see *Figure 1A*). This pattern was evident in both the localizer task and during the main SST.

### β is burst-like in subcortex and cortex

To determine whether β was indeed burst-like in our data, as opposed to an ongoing, oscillatory signature, we used a lagged coherence analysis (as in *Wessel, 2020*). Lagged coherence describes to what degree the current phase of a signal predicts its own phase in the future (*Fransen et al., 2015*).

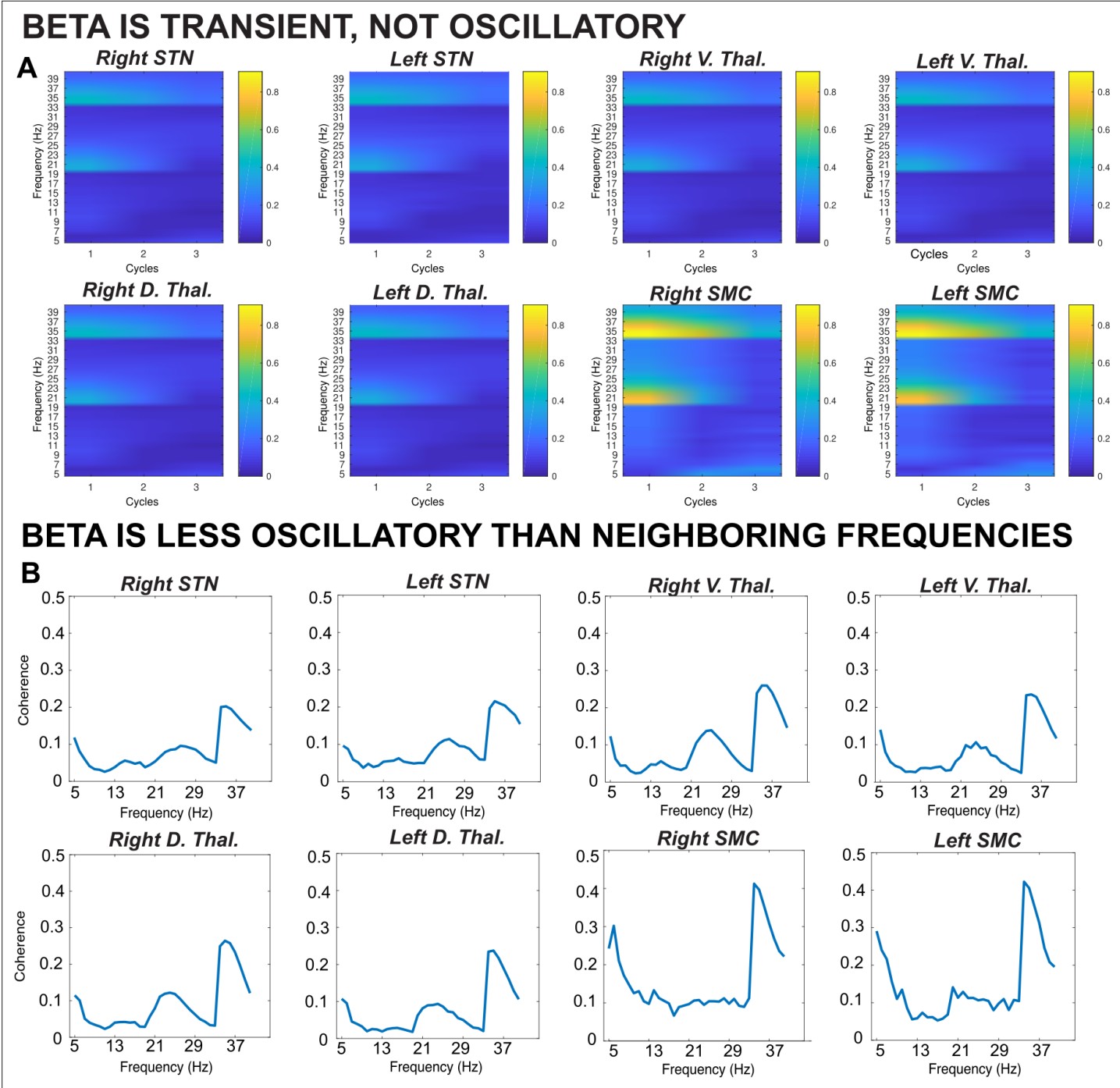

**Figure 2.** Results from a lagged coherence analysis on epoched data including go and stop trials. (**A**) *While coherence is relatively high at one cycle (current phase predicts phase in one cycle well), lagged coherence decreases over time so that lagged coherence in the β band is relatively low at three cycles across recording locations. This supports the claim that β is transient and not oscillatory in nature. (**B**) Coherence shown is at three cycles. The β burst search band (15–29 Hz) represents a relative trough compared to below- and above-β frequencies, indicating that β signals are more transient and less oscillatory than signals in neighboring frequency bands. β is most burst-like in the SMC, slightly less burst-like in STN, and least burst-like compared to neighboring frequencies in the thalamus.*

Signals that are oscillatory in nature can be expected to predict their own activity many cycles on, while phase of a transient signal will be less predictive of future activity in the same frequency band. In line with the assertion that β is a transient and not an oscillatory signal, we observed decreasing levels of lagged coherence in the β band as the number of cycles increased (*Figure 2A*). Moreover, at three cycles, a trough in lagged coherence is observed in the search band for β bursts (15–29 Hz) compared to surrounding frequencies at all recording locations (*Figure 2B*). This lagged coherence trough was most pronounced in the SMC and least pronounced in compared to neighboring frequencies in the thalamus, suggesting that SMC β is most burst-like and thalamic β more sustained in nature. STN β was slightly less burst-like compared to SMC β, but we note this could be an artifact of pathologically long STN β bursts characterized in movement disorders (*Little et al., 2013*; *Anidi et al., 2018*).

Having confirmed that β is indeed burst-like in our recording, we quantified several qualities of the observed β bursts and found that, on average, peak β burst frequency during the burst search window across the entire recording was 22 Hz in STN, 20 Hz in ventral thalamus, 21 Hz in dorsal thalamus, and 22 Hz in SMC. Average β burst duration was 127 ms in STN, 108 ms in ventral thalamus, 111 ms in dorsal thalamus, and 126 ms in SMC. The subsequent analyses described herein address β burst rates and counts, which have been found to be highly predictive of behavior (*Sherman et al., 2016*; *Shin et al., 2017*; *Little et al., 2019*; *Wessel, 2020*; *Jana et al., 2020*; *Enz et al., 2021*).

## β bursts decrease during movement initiation

We first investigated whether the SMC and subcortical sites showed corresponding movement-related reductions in β burst rates leading up to the response (*Wessel, 2020*; *Soh et al., 2021*). Implantation in the motor thalamus specifically targeted the ventral intermediate nucleus (VIM, *Benabid et al., 1993*; *Lozano, 2000*). Neurosurgical implantation of VIM DBS multi-electrode leads positions electrodes on the border of the VIM and the ventral oral posterior (Vop) nuclei of the thalamus, which receive cerebellar (*Na et al., 1997*) and pallidal inputs (*Inase and Tanji, 1995*; *Kuo and Carpenter, 1973*), respectively. Both VIM and Vop are prominently involved in motor function. Due to volume conduction and the small size of these nuclei, we expected the thalamic depth-electrode recordings to be summative recordings from multiple nuclei within motor thalamus (VIM, Vop, and potentially the ventral oral anterior nucleus). To investigate whether contributions of different thalamic nuclei could be parsed in our thalamic recordings, we analyzed the most ventral and most dorsal contact pairs within motor thalamus separately. Notably, the results from both contact pairs were remarkably similar for most statistical comparisons, suggesting that all thalamic contacts captured contributions from a similar set of thalamic motor nuclei. While the same dorsal-ventral distinction is not typically used in analyses of STN (where one specific electrode contact pair can typically be identified as localized in STN by a clear-cut peak in the β activity spectrum), we also performed STN analyses split by dorsal-ventral pairs. However, these analyses are presented in the Figure supplements only. In the main analyses, we follow the convention of selecting one STN contact pair based on the overall amount of detected β bursts throughout the entire recording.

Burst rates were quantified in non-overlapping bins of 100ms starting from the go signal. Indeed, we observed reductions in β burst rates during movement execution in all recording locations. $3 \times 2$ ANOVAs revealed significant effects of *TIMEPOINT* on burst rate in STN ($F_{(2,8)}$ = 8.78, p < 0.0001, $\eta^2$ = 0.31), ventral thalamus ($F_{(2,10)}$ = 6.79, p < 0.0001, $\eta^2$ = 0.25), dorsal thalamus ($F_{(2,10)}$ = 15.85, p < 0.0001, $\eta^2$ = 0.37), and SMC ($F_{(2,19)}$ = 8.04, p < 0.0001, $\eta^2$ = 0.22; see *Figure 1B*). In the STN group, there was also a significant effect of *TRIAL TYPE* ($F_{(2,8)}$ = 4.95, p = 0.02, $\eta^2$ = 0.02) on burst rate, but no *TRIAL TYPE X TIMEPOINT* interaction ($F_{(2,8)}$ = 1.58, p = 0.08, $\eta^2$ = 0.05). On the other hand, for the ventral thalamic electrodes, there was no significant effect of *TRIAL TYPE* ($F_{(2,10)}$ = 2.85, p = 0.08, $\eta^2$ = 0.02) on burst rate, but there was a significant *TRIAL TYPE X TIMEPOINT* interaction ($F_{(2,4)}$ = 1.77, p = 0.04, $\eta^2$ = 0.05). Dorsal thalamic electrodes did not exhibit significant effects of *TRIAL TYPE* ($F_{(2,10)}$ = 0.98, p = 0.39, $\eta^2$ = 0.005) or a *TRIAL X TIMEPOINT* interaction ($F_{(2,10)}$ = 0.68, p = 0.81, $\eta^2$ = 0.02).

For response-locked burst rates, ANOVAs revealed significant effects of *TIMEPOINT* on burst rate in STN ($F_{(2,8)}$ = 4.47, p = 0.0002, $\eta^2$ = 0.27), ventral thalamus ($F_{(2,10)}$ = 3.07, p = 0.005, $\eta^2$ = 0.16), dorsal thalamus ($F_{(2,10)}$ = 2.22, p = 0.03, $\eta^2$ = 0.09), and SMC ($F_{(2,19)}$ = 2.58, p = 0.01, $\eta^2$ = 0.09; see *Figure 1B*). For ventral thalamic sites, there were also significant effects of *TRIAL TYPE* ($F_{(2,10)}$ = 6.92, p = 0.03, $\eta^2$ = 0.01) and a *TRIAL TYPE X TIMEPOINT* interaction ($F_{(2,10)}$ = 2.39, p = 0.02, $\eta^2$ = 0.06).

For dorsal thalamus, there was a significant main effect of *TRIAL TYPE* ($F(2,10)$ = 5.67, p = 0.04, $\eta^2$ = 0.01), but no *TRIAL TYPE X TIMEPOINT* interaction ($F(2,10)$ = 3.07, p = 0.005, $\eta^2$ = 0.03).

These findings are in line with the proposition that β bursts are related to an inhibited state of the motor system, which must be downregulated to achieve a net-disinhibition of the cortico-subcortical motor circuitry and enable movement (e.g. *Soh et al., 2021*).

## β bursts increase during movement cancellation

At rest, the basal ganglia prevent erroneous movement through exertion of tonic inhibition. Furthermore, it has been demonstrated that β bursts are inhibitory with regards to movement (*Little et al., 2019*; *Soh et al., 2021*). Accordingly, the main hypothesis of our study was that after stop-signals in the SST, motor inhibition is achieved by a rapid re-instantiation of an inhibited state in SMC, preceded by β burst signaling from the subcortical nuclei. To test this hypothesis, we first investigated whether successful stop-trials were accompanied by an increase in subcortical β bursting compared to matched go-trials and failed stop-trials.

Indeed, during the critical time period between the onset of the stop signal and the end of SSRT, a significant main effect of *TRIAL TYPE* (successful stop, failed stop, fast and slow matched go) on burst count was found in STN and thalamus (STN: $F(9)$ = 3.89, p = 0.02, $\eta^2$ = 0.30; ventral thalamus: $F(11)$ = 5.89, p = 0.002, $\eta^2$ = 0.35; dorsal thalamus: $F(11)$ = 3.01, p = 0.04, $\eta^2$ = 0.22). Follow-up pairwise *t*-tests revealed that this was due to a significant increase in bursts in the stop signal delay (SSD)-SSRT period for successful stop trials compared to slow matched go trials (STN: $P$ = 0.04; ventral thalamus: p = 0.03; dorsal thalamus: p = 0.03) and successful stop trials compared to failed stop trials (STN: p = 0.04; ventral thalamus: p = 0.04; see *Figure 3*). On the other hand, failed stop trials did not contain significantly more bursts in the SSD-SSRT window compared to fast matched go trials (STN: p = 0.38; ventral thalamus: p = 0.15; dorsal thalamus: p = 0.53). These findings are in line with the assumption that early-latency subcortical β bursting reflects a rapid deployment of inhibitory control after a stop-signal.

As a control analysis, we also compared burst rates across trial types in the baseline periods before the stop signal and go signal. A 3 × 1 ANOVA revealed no significant effects of *TRIAL TYPE* on β bursts in the pre-stop baseline for either STN or thalamus (STN: $F(2,8)$ = 1.09, p = 0.36, $\eta^2$ = 0.11; ventral thalamus: $F(2,10)$ = 2.45, p = 0.11, $\eta^2$ = 0.20; dorsal thalamus: $F(2,10)$ = 1.35, p = 0.28, $\eta^2$ = 0.12). There was also no significant effect of *TRIAL TYPE* on β bursts in the pre-go baseline at any subcortical recording location (STN: $F(2,8)$ = 0.11, p = 0.89, $\eta^2$ = 0.01; ventral thalamus: $F(2,10)$ = 1.09, p = 0.35, $\eta^2$ = 0.10; dorsal thalamus: $F(2,10)$ = 0.25, p = 0.78, $\eta^2$ = 0.02). Hence, the stop-signal related differences in burst rates were not attributable to differences in the baseline rates between trial types.

### Stimulus-locked burst rates

To map out β burst dynamics in the post-go and post-stop periods, we calculated the average burst rate in non-overlapping time bins of 100ms covering the 1000 ms period starting 100ms before the stop-signal (or for matched go trials, the time at which the stop signal would have occurred, as indicated by the current stop-signal delay in the staircase). While this does not take into account each participants' SSRT, it does provide a more comprehensive picture of the development of subcortical β bursting over time. In the STN, we observed significant effects of *TRIAL TYPE* ($F(2,8)$ = 35.14, p < 0.0001, $\eta^2$ = 0.19) and *TIMEPOINT* ($F(2,8)$ = 4.19, p = 0.0003, $\eta^2$ = 0.13), as well as a significant *TRIAL TYPE X TIMEPOINT* interaction ($F(4,8)$ = 5.87, p < 0.0001, $\eta^2$ = 0.14). The same pattern was observed in the ventral thalamus, again with significant main effects of *TRIAL TYPE* ($F(2,10)$ = 26.97, p < 0.0001, $\eta^2$ = 0.20) and *TIMEPOINT* ($F(2,10)$ = 3.16, p = 0.004, $\eta^2$ = 0.08), and a *TRIAL TYPE X TIMEPOINT* interaction ($F(4,10)$ = 6.00, $p$ < 0.0001, $\eta^2$ = 0.15). Moreover, dorsal thalamic electrodes demonstrated significant main effects of *TRIAL TYPE* ($F(2,10)$ = 20.05, p < 0.0001, $\eta^2$ = 0.17) and *TIMEPOINT* ($F(2,10)$ = 3.77, p = 0.001, $\eta^2$ = 0.07), and a *TRIAL TYPE X TIMEPOINT* interaction ($F(4,10)$ = 6.63, p < 0.0001, $\eta^2$ = 0.17).

Pairwise follow-up *t*-tests were used to probe differences between successful stops and go trials and between successful and failed stop trials at individual time bins. Burst rates for successful stop trials were significantly greater than for go trials at 601–700ms (p < 0.001), 701–800ms (p < 0.001), and 801–900ms (p < 0.001) following SSD in the STN and at 501–600ms (p = 0.002), 601–700ms (p = 0.02), 701–800ms (p = 0.02), and 801–900ms (p < 0.001) following SSD in the ventral thalamus (see

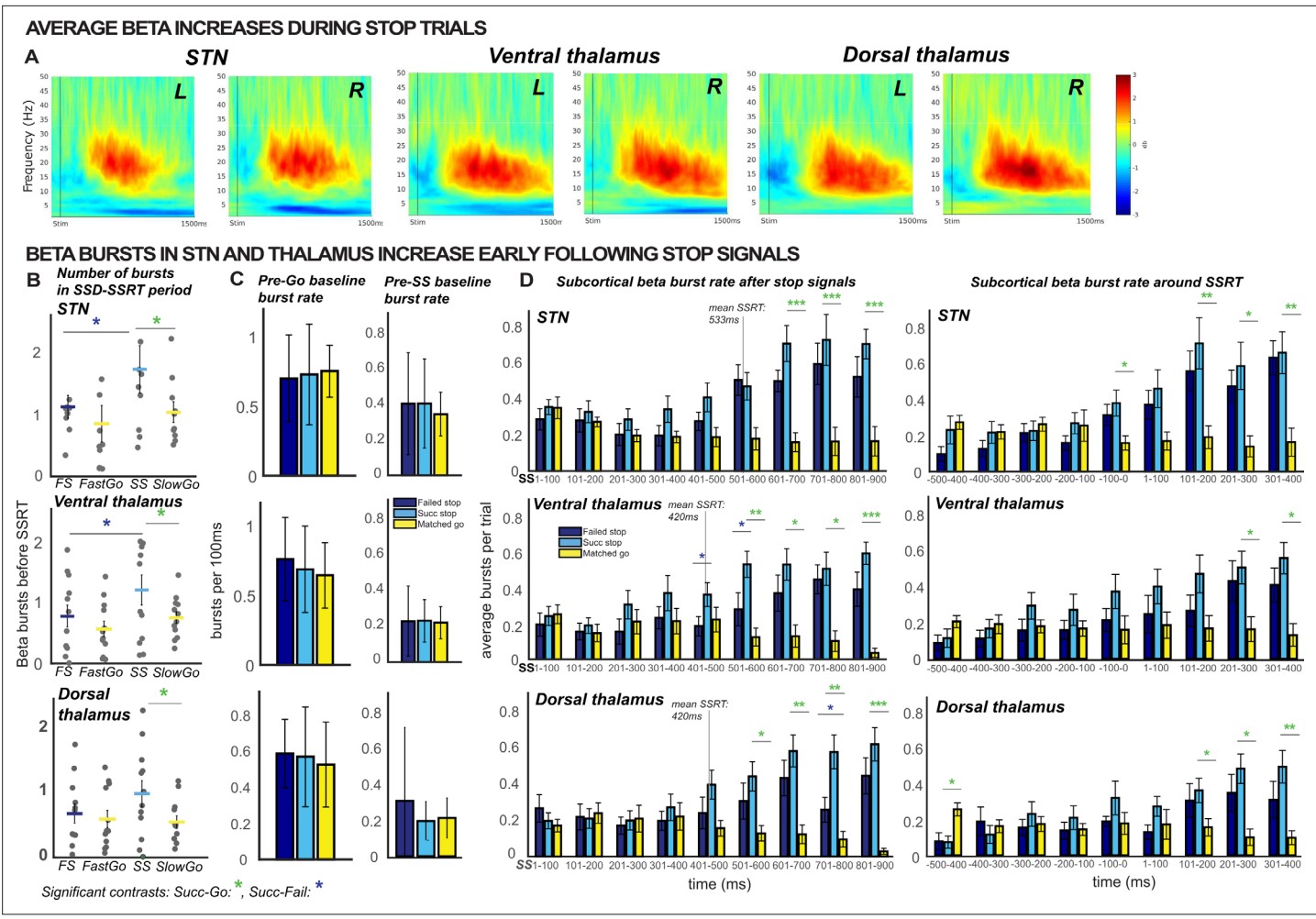

**Figure 3.** β burst rates increase following stop signals. (**A**) Increases in average β band activity are observed at all subcortical recording sites during stop trials (contrast shown is between successful stops and correct go trials). (**B**) The total number of β bursts was quantified between trial-wise stop-signal onset (SSD) and participant-wise SSRT. For matched go-trials, this window began at current SSD stored in the staircase. Each point represents the average burst count for one participant. β bursts increased at early latencies in STN and thalamus during successful cancellation (when quantified between SSD and SSRT) and at later latencies in both thalamus and STN during stop trials (panel D). (**C**) The differences in burst counts and rates in panels B and D could not be accounted for by differences in pre-go or pre-stop baseline burst rates, quantified in the 100ms preceding go- or stop-signal onset. (**D**) Average burst rates at each subcortical recording location time-locked to the stop-signal (left) or subject-wise SSRT (right) are depicted in time bins of 100ms. The gray lines on the time-bin plots show the SSRT sample average for STN and thalamic DBS patient groups. (Significant comparisons key: green stars = comparison between successful stop and go trials, navy stars = comparison between successful and failed stop trial; * indicates p < 0.05, ** indicates p < 0.01, *** indicates p < 0.001. Significant effects displayed in the plot are effects of TRIAL TYPE, on burst counts (**B**) or on burst rates at given time points (**D**).

The online version of this article includes the following figure supplement(s) for figure 3:

**Figure supplement 1.** Participants performed an auditory version of the stop-signal task while LFP recordings were collected.

**Figure supplement 2.** β burst rates increase following stop signals in ventral and dorsal STN.

**Figure supplement 3.** Comparing β burst counts and rates across hemispheres.

**Figure supplement 4.** Exemplar estimations of recording lead locations based on pre- and post-operative imaging and planned implant trajectory in a VIM and STN DBS case.

*Figure 3*). In the ventral thalamus, there were also significant differences between successful and failed stop trial burst rates at 401–500 (p = 0.04) and 501–600ms (p = 0.05) following SSD. In dorsal thalamus, significant differences in burst rates were observed between successful stops and go trials at 501–600 (p = 0.02), 601–700 (p = 0.004), 701–800 (p = 0.002), and 801–900ms (p < 0.001), and between successful and failed stop trials at 701–800ms (p = 0.01). Note that there was no increase in

pre-SSRT β bursting on successful stop-trials in this bin-wise quantification, which is at odds with the above-mentioned quantification that measured β burst counts in each individual subjects' stop-signal-to-SSRT period. This suggests a tight relationship between β burst dynamics and inhibitory control behavior measured by each individual's SSRT – resulting in the fact that differences in β burst rates after stop-signals are obscured when between-subject differences in SSRT are not taken into account. In line with the hypothesis that β bursts reflect the re-instantiation of tonic rest inhibition, burst rates in the first bin with a significant difference between successful stop-go trial were not significantly different from burst rates in the pre-go baseline (STN: p = 0.97; ventral thalamus: p = 0.43; dorsal thalamus: p = 0.13), during which tonic inhibition is present. However, there were significant differences between burst rates in these time bins and the pre-stop baseline in the thalamus (ventral thalamus: p = 0.003; dorsal thalamus: p = 0.008), but not the STN (p = 0.08).

### SSRT-locked burst rates

In the STN, we observed significant effects of TRIAL TYPE ($F(2,8) = 25.62$, $p < 0.0001$, $\eta^2 = .11$) and TIMEPOINT ($F(2,8) = 5.25$, $p < 0.0001$, $\eta^2 = 0.17$), as well as a significant TRIAL TYPE X TIMEPOINT interaction ($F(4,8) = 6.99$, $p < 0.0001$, $\eta^2 = 0.17$) on burst rates in time bins locked to participant-wise SSRT. The same pattern was observed in the ventral thalamus, again with significant main effects of TRIAL TYPE ($F(2,10) = 8.57$, $p = 0.002$, $\eta^2 = 0.12$) and TIMEPOINT ($F(2,10) = 5.51$, $p < 0.0001$, $\eta^2 = 0.13$), and a TRIAL TYPE X TIMEPOINT interaction ($F(4,10) = 3.94$, $p < 0.0001$, $\eta^2 = 0.11$). Moreover, dorsal thalamic electrodes demonstrated significant main effects of TRIAL TYPE ($F(2,10) = 5.96$, $p = 0.009$, $\eta^2 = 0.07$) and TIMEPOINT ($F(2,10) = 4.89$, $p < 0.0001$, $\eta^2 = 0.09$), and a TRIAL TYPE X TIMEPOINT interaction ($F(4,10) = 4.56$, $p < 0.0001$, $\eta^2 = 0.15$).

Pairwise follow-up *t*-tests were used to probe differences between successful stops and go trials and between successful and failed stop trials at individual time bins surrounding SSRT. In the STN, significant differences between β burst rates during successful stop and go trials were observed at 100–0ms (p = 0.047) preceding the stop-signal and at 101–200ms (p = 0.009), 201–300ms (p = 0.04), and 301–400ms (p = 0.004) following the stop-signal. In the ventral thalamus, significant differences between β burst rates during successful stop and go trials were observed at 201–300ms (p = 0.047) and 301–400ms (p = 0.04) following the stop-signal. In the dorsal thalamus, significant differences between β burst rates during successful stop and go trials were observed at 500–400ms (p = 0.02) before the stop-signal and at 101–200ms (p = 0.02), 201–300ms (p = 0.01), and 301–400ms (p = 0.006) following the stop-signal.

## STN β bursts upregulate SMC bursts during cancellation

The proposition that inhibitory control is implemented via a rapid re-instantiation of SMC inhibition following β bursts in STN/Thalamus implies that SMC β burst rates should be increased in the immediate aftermath of subcortical bursts (cf., *Wessel, 2020* for a demonstration of the same relationship between β bursts at fronto-central scalp sites likely reflecting cortical regions of the stopping network upstream from the subcortical nuclei investigated here and subsequent β-burst rates over SMC). To test this, we quantified SMC β bursting time-locked to the first subcortical β bursts following the stop-signal within 500ms. Indeed, there was a main effect of TIMEPOINT ($F(2,8) = 5.22$, $p = 0.006$, $\eta^2 = 0.14$) on SMC bursts time-locked to STN bursts. Moreover, follow-up *t*-tests revealed that bursts in STN were followed within 50ms by a difference between burst rates for successful stops compared to go trials, with burst rates increasing for successful stops, though this difference did not survive multiple-comparisons corrections (uncorrected p = 0.03; see *Figure 4*).

Our 2 × 2 ANOVA did not reveal a significant effect of TRIAL TYPE ($F(2,8) = 0.54$, $p = 0.59$, $\eta^2 = 0.01$) or a TRIAL TYPE x TIMEPOINT interaction ($F(4,8) = 1.94$, $p = 0.09$, $\eta^2 = 0.07$) for SMC bursts time-locked to STN bursts. Likewise, no effects of TRIAL TYPE ($F(2,10) = 1.55$, $p = 0.24$, $\eta^2 = 0.05$), TIMEPOINT ($F(2,10) = 1.71$, $p = 0.19$, $\eta^2 = 0.03$), or an interaction ($F(4,10) = 0.81$, $p = 0.58$, $\eta^2 = 0.04$) were found for SMC bursts time-locked to ventral thalamic bursts. There was a significant effect of TIMEPOINT ($F(2,10) = 4.15$, $p = 0.01$, $\eta^2 = 0.09$) on burst rates in SMC following dorsal thalamic bursts, but no effect of TRIAL TYPE ($F(2,10) = 0.31$, $p = 0.74$, $\eta^2 = 0.01$) or a TRIAL TYPE x TIMEPOINT interaction ($F(4,8) = 0.15$, $p = 0.99$, $\eta^2 = 0.005$). There were no significant pairwise differences between burst rates in ventral or dorsal thalamus for successful stops versus failed go trials. As described in the previous paragraph, despite the absence of a significant main effect of TRIAL TYPE, we conducted

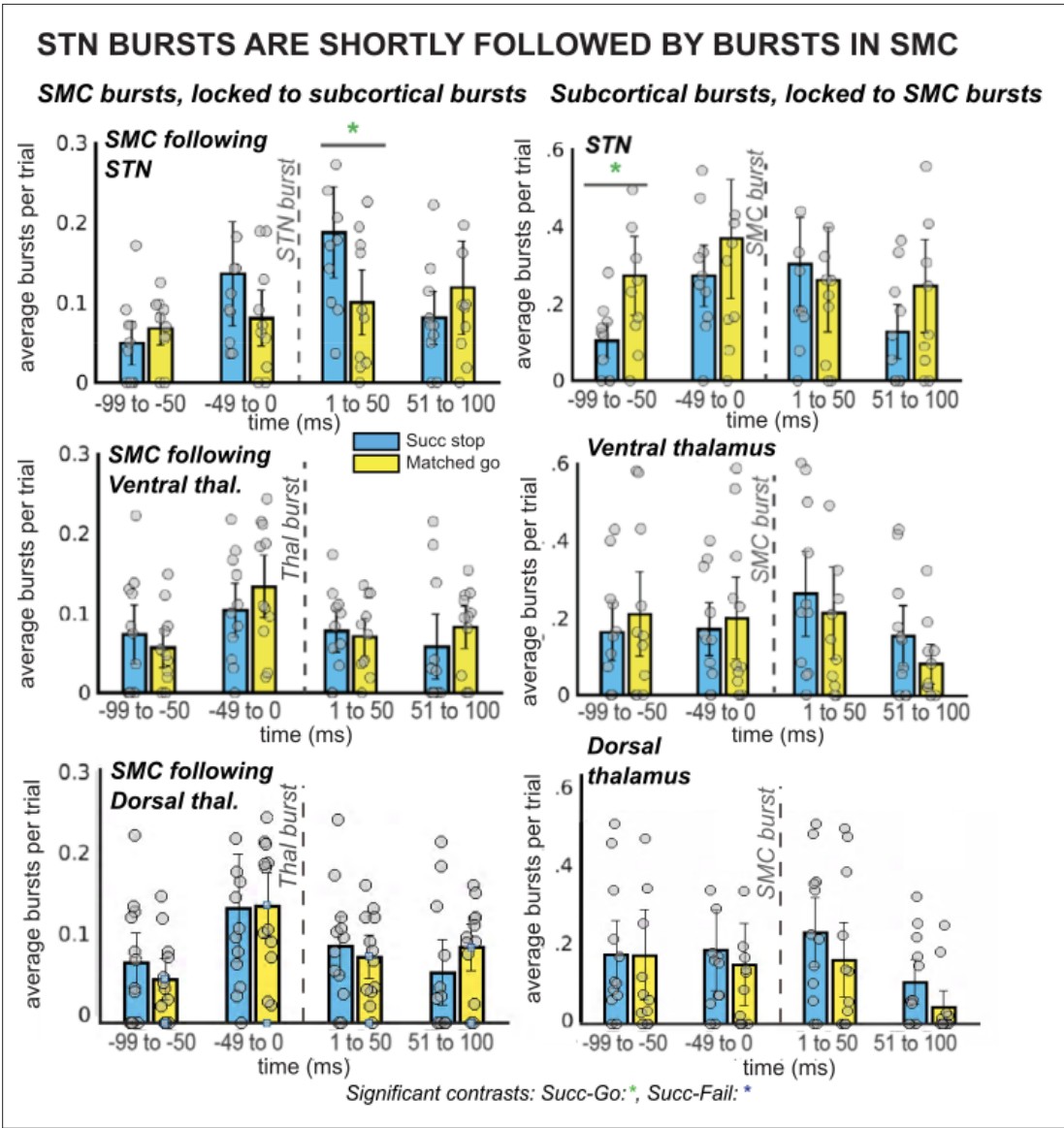

**Figure 4.** β burst rates in SMC increase following STN bursts. The left column of plots displays the timing of SMC bursts with respect to subcortical bursts, in 50ms time bins. The right column displays timing of subcortical bursts with respect to SMC bursts. All burst rates were calculated in the 500ms following stop-signal onset (or SSD, for matched-go trials). Each gray dot is the average burst rate per participant in the given time bin. In the STN group, the first STN burst following the stop signal (or SSD, for matched-go trials) was followed within 50ms by bursts in the SMC for stop, but not go, trials. This reliable temporal relationship between STN and SMC bursts during movement cancellation did not follow the opposite pattern – STN bursts did not reliably follow SMC bursts at a specific time point. (Significant comparison key: green stars = comparison between successful stop and go trials; * indicates uncorrected p < 0.05. Effects indicated in the figure are effects of TRIAL TYPE on burst rate at given time points.)

The online version of this article includes the following figure supplement(s) for figure 4:

**Figure supplement 1.** β burst rates in SMC preceding and following bursts in ventral and dorsal STN.

pairwise tests between successful go and matched go burst rates at individual time points because of our strong a priori hypothesis that β bursts in SMC would increase following STN bursts. This hypothesis was derived directly from a similar observation in *Wessel, 2020*, wherein SMC β bursts were increased within 25ms of frontocentral β bursts.

Conversely to the increase in SMC burst rate after STN bursts, we did not see an increase of STN β bursts *following* SMC bursts. A 2 × 2 ANOVA of burst rates in STN, ventral thalamus, and dorsal thalamus time-locked to SMC bursts revealed an effect of *TIMEPOINT* in all regions (STN: $F(2,8) = 3.73$, p = 0.02, $\eta^2 = 0.08$; ventral thalamus: $F(2,10) = 3.10$, p = 0.04, $\eta^2 = 0.05$; dorsal thalamus: $F(2,10)$

= 2.89, p = 0.05, $\eta^2$ = 0.04) and an effect of TRIAL TYPE in ventral thalamus ($F(2,10)$ = 5.32, p = 0.01, $\eta^2$ = 0.13), but no significant main effects of *TRIAL TYPE* in STN ($F(2,8)$ = 0.76, p = 0.48, $\eta^2$ = 0.03) or interactions between the two factors (STN: $F(4,8)$ = 0.69, p = 0.66, $\eta^2$ = 0.03; ventral thalamus: $F(4,10)$ = 1.12, p = 0.36, $\eta^2$ = 0.04). In dorsal thalamus, there were no main effects of *TRIAL TYPE* ($F(2,10)$ = 0.21, p = 0.81, $\eta^2$ = 0.01) or a *TRIAL TYPE x TIMEPOINT* interaction ($F(4,8)$ = 0.94, p = 0.47, $\eta^2$ = 0.02). No significant pairwise comparisons were found for burst rates during successful stops compared to go trials at individual time points in any region except for a difference at the –100 to –50ms timepoint in the STN was significant before multiple comparisons correction (uncorrected p = 0.02).

The observation of elevated SMC β bursts following, but not preceding, STN bursts supports the proposition that subcortical bursts lead to a rapid upregulation of SMC bursts during stopping.

### STN β bursts precede thalamic bursts during cancellation

A key prediction of existing network models of inhibitory control is that STN is upstream from motor thalamus (specifically, from the pallidal projection regions in Vop). In other words, during the purported cascade that results in movement cancellation, STN signaling should temporally precede thalamic signals. To test whether this is the case for the β burst signals observed in this study, we calculated the average latency of the first β burst after the stop signal for each subcortical recording site and compared them between groups (as well as within a single subject with simultaneous recordings from both sites). To account for differences in SSRT across participants, we quantified the onset latency of first bursts with respect to participant-wise SSRT. Across the group-level sample, STN bursts on average occurred before ventral and dorsal thalamic bursts during stop trials (see *Figure 5A*). While there was a significant effect of first burst time on TRIAL TYPE (with the first burst occurring earlier on successful stop trials compared to failed stop-trials in both ventral thalamus and STN; $F(1,19)$ = 8.32; p = 0.01; $\eta^2$ = 0.02), there was no significant effect of *LOCATION* on average burst timing ($F(1,19)$ = 1.90; p = 0.18; $\eta^2$ = 0.08), and no interaction ($F(1,19)$ = 0.42; p = 0.52; $\eta^2$ = 0.001) when comparing STN to ventral thalamic contacts specifically. However, a 2 × 2 between and within-factors ANOVA revealed a significant *TRIAL TYPE X LOCATION* interaction for burst timing in *STN and dorsal thalamus* ($F(1,19)$ = 5.06; p = 0.04; $\eta^2$ = 0.02), although main effects of *TRIAL TYPE* ($F(1,19)$ = 1.50; p = 0.24; $\eta^2$ = 0.005) and *LOCATION* ($F(1,19)$ = 2.98; p = 0.10; $\eta^2$ = 0.12) were not present. Moreover, a follow-up *t*-test revealed significant differences between timing of STN and dorsal thalamic β bursts with respect to SSRT during successful stops (p = 0.03).

However, the ultimate test of burst timing differences across regions is provided by the single subject who had recordings from both regions, as this provides the only comparison performed in a situation with identical behavior (specifically, equal SSRT, which in this subject was 304ms). In line with the qualitative pattern observed in the group-level comparison, in this single subject, bursts in the STN occurred significantly earlier than bursts in the thalamus (specifically, in the dorsal thalamus) during stop trials with respect to the stop-signal (see *Figure 5B*), with *t*-tests between recording regions revealing significant differences between burst timing in STN and dorsal thalamus during successful stop trials (p = 0.03, *one-sided)*, but not during failed stop trials (p = 0.19, *one-sided*). (These tests are one-tailed because of the strong a priori hypothesis that STN bursts would precede thalamus, and not the other way around.) Though some single-trial bursts across recording locations occurred *after* subject SSRT, the median burst timing occurred prior to SSRT in STN and ventral thalamus. The majority of STN and ventral thalamic bursts occurred before SSRT as well (see *Figure 5B*). This observation that cancellation-related STN bursts occur before bursts in the thalamus also supports accounts that movement regulation may be accomplished by STN-facilitated inhibition of the thalamus during a period before behavioral cancellation is observed (i.e., before SSRT).

## Discussion

We used simultaneous, multi-site intracranial recordings in awake, behaving humans to delineate the cortico-subcortical β-burst dynamics that underlie the inhibitory control of movement. Our findings have significant implications for our understanding of inhibitory control in the human brain, as well as the nature of β signaling in human motor circuitry.

# STN BURSTS PRECEDE THALAMIC BURSTS IN STUDY AVERAGE

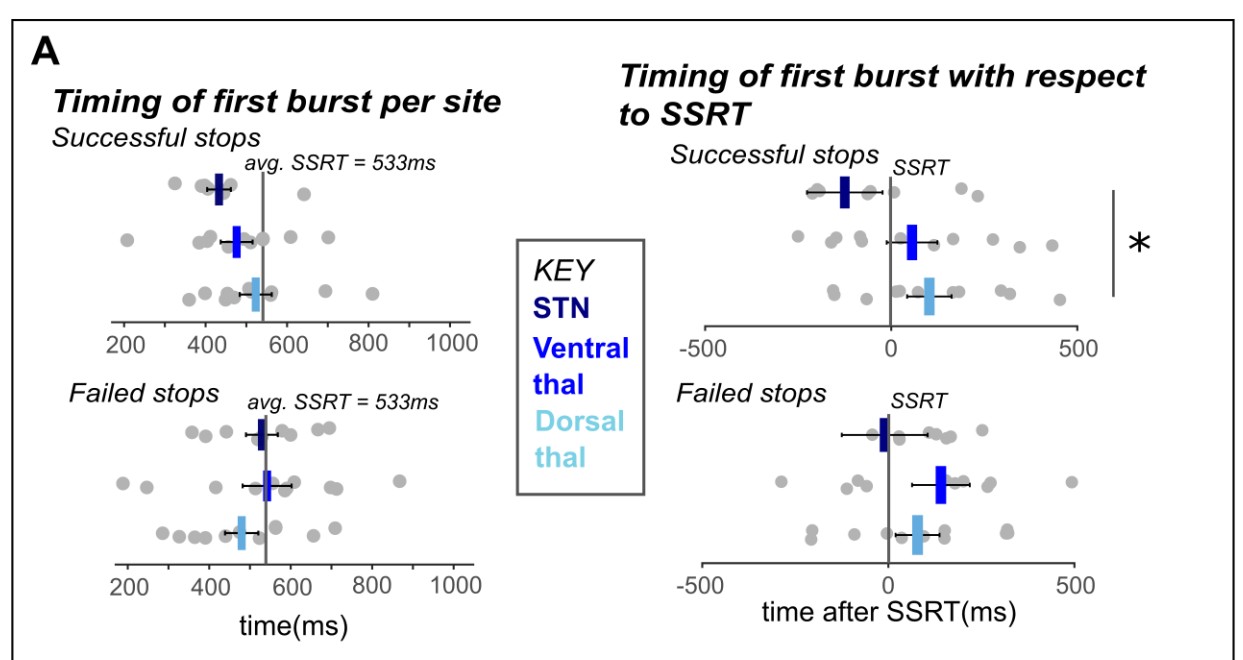

## STN BURSTS PRECEDE THALAMIC BURSTS AT SINGLE SUBJECT LEVEL

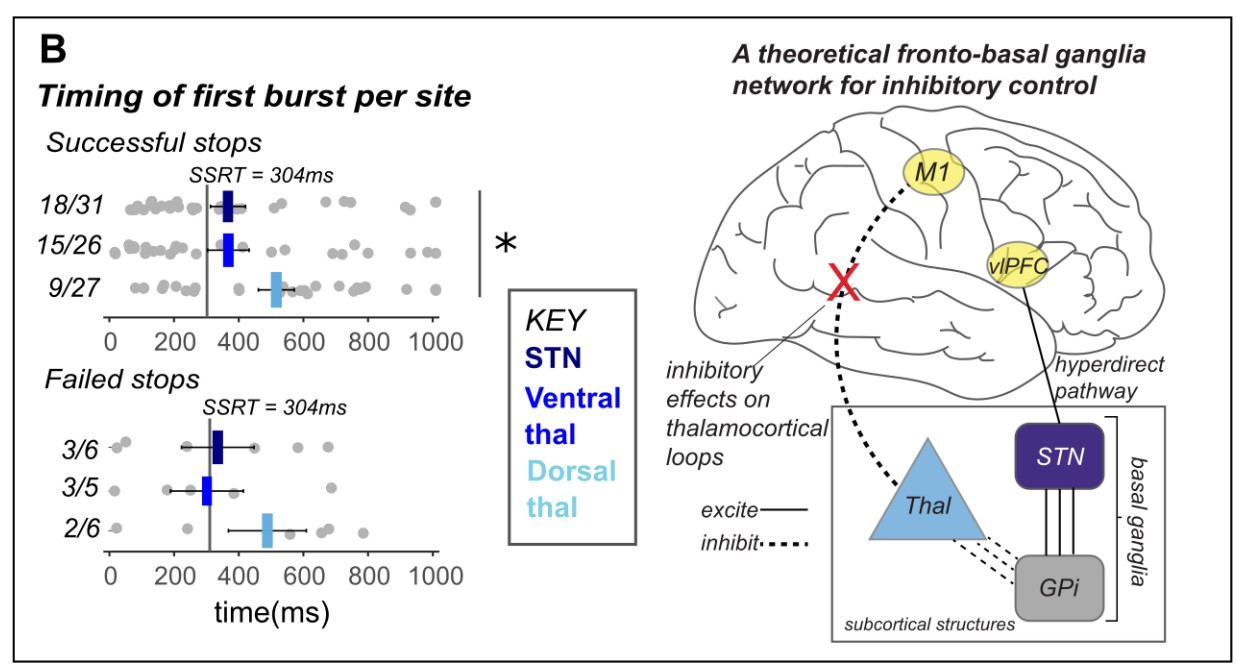

**Figure 5.** *β bursts in STN precede bursts in thalamus.* **A**) *The timing of first bursts from the STN in STN DBS patients and thalamic regions in thalamus DBS patients are shown with respect to the stop-signal (left) and participant-wise SSRT (right). First bursts were quantified between stop-signal onset and 1 s following stop-signal onset. Each gray dot represents a participant's mean burst timing for each trial type and recording location. The central line on the barplots represents the average of single-subject mean burst timings.* (**B**) *The timing of first bursts from the STN and thalamic regions in the single subject with both STN and thalamic DBS are shown with respect to the stop-signal (left) and participant-wise SSRT (right). Each dot represents the timing of a burst for a single trial, while the central bar represents the median burst timing. Bursts that occur in each region before SSRT are counted. Across the entire study sample (**A**) and in our single subject with simultaneous STN and thalamic recording sites (**B**), STN bursts occurred earlier than dorsal thalamic bursts during cancellation. These findings from subcortical regions in our datasets lend support for an account of subcortical dynamics proposed in a theorized network model of movement cancellation, which posits that the STN is recruited prior to and acts to net-inhibit the thalamus*

*Figure 5 continued on next page*

Figure 5 continued

during cancellation (red X indicates reduction of thalamocortical drive). (Significant comparison key: stars = comparison between burst timing in STN and dorsal thalamus; * indicates p < 0.05. Effects indicated in the figure are effects of BURST LOCATION on average burst timing.)

### β bursts in the STN and thalamus relate to movement cancellation

β bursts in the human STN and thalamus relate to the rapid deployment of inhibitory control during movement cancellation in the SST. An analysis of subcortical β burst counts between SSD and SSRT revealed a greater number of bursts present during successful stops than during failed stops or matched go trials. Moreover, the analysis of first-burst timing revealed that in both STN and thalamus, the timing of the first burst distinguished successful from failed stopping, as earlier bursts lead to significantly higher rates of stopping. This is also prominently in line with the horse race model of the stop-signal task (*Logan, 1983*). (This finding of differential activation in STN depending on the success of stopping appears in conflict with recent 7T fMRI work that found greater BOLD activity in STN for failed compared to successful stops [*Miletić et al., 2020*]. However, the differences between these findings likely reflect the respective data collection methods [with fMRI having less temporal resolution] and the fact that action errors also activate STN [*Cavanagh et al., 2014*; *Wessel and Aron, 2017*]. Hence, we tentatively propose that greater BOLD activation in STN for failed stops reflects an additive effect of a double-activation of STN on failed stop-trials, where the initial stop-related activity and the subsequent error-related activity is summed up, whereas the latter activity is absent on successful stop-trials.)

Though there are inherent limitations to the ability to record from specific nuclei within the human motor thalamus, even when using stereotactic depth electrode recordings, the findings of elevated β bursting for successful stop trials within ventral thalamic electrodes supports the proposition that activity in motor thalamus is involved in movement cancellation. Our findings in intraoperative movement disorder participants mirror results recently obtained from scalp recordings in healthy subjects (*Wessel, 2020*, *Jana et al., 2020*), which demonstrated increased and earlier β bursts during movement cancellation at cortical sites that are ostensibly up-stream of the basal ganglia circuitry investigated here (cf., *Chen et al., 2020*). The current study is the first to concretely demonstrate the relationship between subcortical β bursts across multiple subcortical regions during movement cancellation in the human brain.

In addition to this early-latency β bursting in the SSD-SSRT period, we also observed clear increases in β bursts at later latencies, clearly after SSRT, in both subcortical regions. While we had no hypothesis about such a finding a priori (as the *Mosher et al., 2020* report of these late-latency β bursts was published after our investigation concluded), we surmise that these later peaks in β bursting may relate to the slower activation of the basal ganglia indirect pathway during movement cancellation (*Jahfari et al., 2011*; *Sano et al., 2013*; *Schmidt et al., 2013*; *Mallet et al., 2016*). Indeed, one recent framework of movement cancellation, supported by a body of neurophysiological work in rodents, contains the proposition that stopping is a two-step and not a unitary process (*Schmidt et al., 2013*; *Schmidt and Berke, 2017*; *Diesburg and Wessel, 2021*). This two-step model, termed the 'Pause then Cancel' model, consists of two phases. An initial Pause involves rapid gating of STN through hyperdirect pathway activation. Meanwhile, a parallel Cancel phase implements indirect pathway activation to eliminate drive to movement from the direct pathway. The STN plays a critical role during both of these phases. Indeed, computational modeling of the human basal ganglia has suggested that inhibitory control could rely on activation of both the hyperdirect and indirect basal ganglia pathways in parallel. While the former implements the rapid gating of the STN, thereby raising the response threshold, the latter ultimately removes drive to movement from the direct basal ganglia pathway (*Frank, 2006*; *Wiecki and Frank, 2013*). If β bursts do in fact relate to both hyperdirect and indirect pathway activation, this has major implications for emerging theories of movement cancellation. Within the current dataset, the subcortical β burst dynamics are in line with this proposition, making them a candidate signature of a unitary signal that coordinates both hyperdirect and indirect pathway inhibition.

### STN β bursts may influence motor output by raising SMC burst rates

By leveraging unique simultaneous recordings of SMC and subcortical regions, we found that early bursts in STN following stop signals appear to influence motor output. Indeed, the β burst rates in SMC

were increased significantly within about 50ms after cancellation-related bursts in STN. Conversely, there was no such increase in STN bursting following SMC bursts (nor, surprisingly, in motor thalamus, in either direction). This suggests a degree of directionality in the relationship between bursts across these regions. Because of the specific directionality of this finding, it suggests that β bursts in STN might influence β burst rates in SMC. Given that β bursts in SMC have an inhibitory influence on motor output (*Little et al., 2019*), this points to a possible mechanism by which the STN can influence motor output during movement cancellation, though further research would be required to confirm a causal relationship.

## STN β bursts precede thalamic bursts during movement cancellation

In line with the proposed cortico-subcortical cascade underlying inhibitory control, according to which cortical signals to STN lead to inhibition of the thalamus via the GPi, we found that β bursts following the stop signal across our study sample occur in STN earlier on average than in the motor thalamus. Specifically, STN bursts preceded bursts in the dorsal thalamus by a significant delay during successful stops. (Concerns about volume conduction and nuclei size in the motor thalamus notwithstanding, it is worth noting that these dorsal-most contacts are the ones most likely to be located in Vop according to the trajectory of implantation [see Materials and methods section] – i.e., the region that receives ostensibly inhibitory pallidal projections as part of the inhibitory basal ganglia pathways [see below].) A subsequent assessment of subcortical burst latency in the patient with both STN and thalamic recordings confirmed that stop signal-related STN β bursts on average occur prior to dorsal thalamic bursts within the same individual. Notably, this finding was a result of a highly unusual case, wherein thalamic implants were revised and STN implants placed, lending an opportunity to record from both regions at once. This evidence lends insight into human inhibitory control circuits that would be impossible to obtain under any other circumstances or by using nonin-vasive methodological approaches in humans. We chose to quantify the timing of first post-stop-signal bursts in subcortical regions but not SMC because there are baseline levels of non-evoked β bursting in SMC (which are further increased in proactive control contexts, i.e., *Soh et al., 2021*) which make quantifying timing of first evoked bursts in SMC difficult. To investigate the relationship between subcortical and cortical bursts we used burst-locked analyses (*Figure 4*) that are more likely to quantify evoked bursts.

Though our simultaneous recordings from STN and motor thalamus provide unique insights rarely obtained in human neuroscience research, our insights into the exact nuclei of the thalamus that produce the β bursts observed here or provide proximal or distal drives to neocortical neurons during sensorimotor bursts (c.f. *Sherman et al., 2016*), are somewhat limited. Notably, the VIM receives inputs from cerebellum and shows strong connectivity to the primary motor cortex (*Klein et al., 2012*). Hence, while this nucleus is well-positioned to influence activity in motor cortex, it is unlikely it would receive inhibitory inputs from the pallidum. On the other hand, Vop receives pallidal efferents and projects broadly to motor-related frontal cortices (mostly SMA, but also to premotor and M1; *Sakai et al., 1999*; *Hyam et al., 2012*). Our recordings likely include contributions from both nuclei, given that both dorsal and ventral thalamic contacts showed qualitatively and quantitatively similar patterns throughout the study. In regard to whether either nucleus participates in the thalamic drives that ostensibly generate sensorimotor β bursts (*Sherman et al., 2016*; *Shin et al., 2017*), a cycle-by-cycle analysis of the waveform underlying subcortical β bursts might provide insight into whether the progression of β bursts throughout regions of the basal-ganglia-thalamic circuit reflects propagation of a morphologically identical inhibitory signal.

While our results support the view that β bursts signify motor inhibition at the level of the basal ganglia, thalamus, and SMC, we did not find the same systematic temporal relationship between β bursts in thalamus and in SMC that was found for STN and SMC – that is, the ostensible subcortical start- and cortical endpoints of the cascade. While somewhat surprising, this is in line with existing work showing that thalamocortical motor representations may depend on neural interactions outside of the β band (*Opri et al., 2019*). Therefore, it is possible that the subcortico-cortical cascade of processing underlying motor inhibition starts with burst-like β signals from STN to thalamus and ends with the re-emergence of β bursts in SMC, but that the inhibition of motor activity between thalamus and SMC does not itself involve β bursts. In this scenario, the tight temporal relationship between the emergence of β in STN and SMC during stopping would be merely an indirect effect of the fact that

β bursts signify the rapid reinstantiation of inhibition at both levels, rather than directly reflecting the propagation of β commands through the entire basal ganglia-thalamus-SMC chain.

## Broad and clinical implications

Here, we present evidence of an association between β bursts and movement cancellation across a cortico-subcortical FBg pathway for inhibitory control. This establishes transient β as a candidate signature of inhibitory commands in fronto-basal ganglia motor circuits, in line with recent proposals regarding the inhibitory neuronal processes reflected in such β bursts (*Sherman et al., 2016*; *Shin et al., 2017*). β bursts carry information about motor output on the single-trial level, therefore providing a powerful window into the nature and timing of motor control in the brain.

Notably, these findings also hold relevance for emerging clinical treatments of movement disorders. It is still not well understood how continuous DBS modulates local circuits and network-level activity to change control-related behavior, and studies on this topic have produced mixed findings. For example, in some cases, STN DBS reduces SSRT (*Mirabella et al., 2011*; *Roy et al., 2020*), while some studies have observed SSRT increases instead (*Ray et al., 2009*; *Obeso et al., 2013*). A better understanding of subcortical control circuits, such as understanding that the STN precedes thalamic recruitment during movement cancellation, may improve our insights into why DBS produces such mixed effects on control-associated behavior. Moreover, such an improved understanding of those subcortical connections may also shed light on why movement disorder neuropathologies bring about deficits in inhibitory control more generally. In addition, adaptive deep brain stimulation (aDBS) is an intervention being tested for the treatment of motor symptoms in movement disorders such as PD. This type of DBS suppresses abnormally high levels of β power in STN by stimulating only when sensors are activated by long, high amplitude β bursts (*Little et al., 2013*; *Little et al., 2016*; *Meidahl et al., 2017*). Although recent testing of aDBS suggests that these stimulation protocols work by curtailing pathological and not physiological bursts (*Anidi et al., 2018*) – and may in fact spare more physiological β than continuous DBS (*Tinkhauser et al., 2017*)– no studies have investigated the effects of aDBS on inhibitory control processes. More research of aDBS effects on physiological bursts during inhibitory control would rule out potential complications for control processes and provide the opportunity to evaluate the effects of physiological and pathological β bursts on network-wide β bursts and behavior.

## Limitations

Some additional limitations to the current work, beyond the likely volume conduction in the thalamic recordings, are worth mentioning. Notably, our current finding is – superficially – at odds with a recent report from Mosher and colleagues (*Mosher et al., 2020*), who purportedly showed that human STN and SMC β bursts are dissociated from activity in movement-associated neurons. Specifically, while in their data, movement-related neurons in STN showed reduced firing prior to movement cancellation during successful stops, β bursts were not observed until later latencies. However, while we here replicated the later-latency β bursting, the apparent discrepancy in the early-trial β burst rates is likely a reflection of differences in how β bursts were measured following the stop-signal. Specifically, Mosher and colleagues quantified β burst rates in a continuous manner after the stop signal, using sliding windows (similar to what was done in *Figure 1B* in the current manuscript). However, since SSRT varies considerably across participants (particularly in clinical samples), this procedure does not take the substantial between-subject variance in SSRT into account. Indeed, while the stop-signal locked bin-quantification in our study also didn't show significant increases in burst rates between stop and go trials, quantifying β bursts in each individual's SSD-to-SSRT period revealed clear differences.

Other recent work has also put forth the argument that β bursts – at least in scalp recordings – do not occur regularly enough to index the deployment of inhibitory control during movement cancellation (*Errington et al., 2020*). We stress that though β bursts may not occur on every single successful stop trial in humans, our current intracranial data show at least one STN β burst on average during the SSD-SSRT interval (see *Figure 2*, first panel). Moreover, we used a very conservative amplitude threshold for burst identification (adapted from one of the landmark investigations of β bursts in humans, *Shin et al., 2017*), which limited the number of β bursts analyzed. Future methodological developments may enable researchers to use more adaptive thresholding procedures (as in *Enz et al., 2021*), which may reveal that actual β burst rates are perhaps higher than what is commonly observed.

Finally, the use of a patient population was necessary to obtain deep brain recordings, but consequentially these findings may not generalize entirely to healthy individuals. Our participants were undergoing brain surgery at the time they performed the behavioral task, which may have caused distraction or fatigue. At the beginning of the surgery, patients were administered dexmedetomidine, an intravenous sedative. Though patients were required to be awake and responding to instructions from the clinical team at least 30 min before our recordings and off all sedatives, we cannot rule out lingering effects of sedation. These limitations are endemic to the endeavor of human intracranial neurophysiology. A related limitation is the possibility that some β bursts in our data may be non-representative due to the higher number of pathologically long, high-amplitude β bursts identified in patients with movement disorders, especially PD (*Tinkhauser et al., 2017*; *Lofredi et al., 2019*). However, the use of a high cut-off amplitude threshold to identify β bursts (such as the one we used here) has been shown to bias burst selection in favor of shorter-duration β bursts (*Schmidt et al., 2020*), suggesting that our quantification methods might have been more likely to sample the shorter (though, still high-amplitude) β bursts.

## Conclusion

In conclusion, this study provides network-level neurophysiological evidence for a proposed cascade of cortico-subcortical processing, during which β band burst-like signals between STN, thalamus, and SMC are related to inhibition of motor output. This was achieved using a highly unique sample of multi-site intracranial recordings – including a simultaneous recording from both subcortical sites – that is unprecedented in human cognitive neuroscience studies. We found that both STN and thalamus showed increased β burst signaling in the critical time period following the stop-signal, and that STN bursts in particular were followed at low latency by β bursting in SMC. Given that these SMC bursts have been associated with an inhibited state of the motor system (*Little et al., 2019*; *Soh et al., 2021*) this strongly speaks in favor of the theory that action stopping is achieved via a rapid re-instantiation of inhibitory control following β burst signaling from the subcortical basal ganglia. In addition, β bursts in STN temporally preceded bursts in thalamus in a single-subject case, lending preliminary support for circuit models of inhibitory control which propose that inhibitory STN activity precedes activity in the thalamus. These findings further confirm transient β bursts as a signature of inhibitory control in fronto-basal ganglia circuits of the human brain.

# Materials and methods

## Participants

Twenty-three adult participants were recruited at the University of Iowa Hospitals and Clinics from all neurosurgical candidates slated for DBS electrode implantation in the thalamus (specifically targeting the ventral intermediate nucleus, VIM) or STN. STN DBS patients had a diagnosis of idiopathic PD and thalamic DBS patients had a diagnosis of essential tremor. One patient, with a diagnosis of PD, had existing thalamic implants revised and STN implants placed within the same surgery. From this patient, we recorded data from unilateral thalamus and bilateral STN. Administration of dopaminergic medication was withheld for over 8 hr before DBS surgery for all patients. Two participants' data were excluded from analyses based on behavioral performance, leaving a sample of 21 participants (nine female, mean age: 67 years, age range: 52–78). Information regarding handedness, symptom laterality, and motor symptom severity for participants included in analyses can be found in the table included in *Supplementary file 1*. When available, pre-operative Unified Parkinson's disease rating scale (UPDRS; *Movement Disorder Society Task Force on Rating Scales for Parkinson's Disease, 2003*) part III motor examination total scores are included for PD patient participants and Fahn-Tolosa tremor scale scores (*Fahn et al., 1993*) are included for essential tremor patient participants. UPDRS scores are presented as totals of 33 scored items with possible scores of 0–4, and Fahn-Tolosa scores are presented as a total of 21 scored items, with a possible score of 0–4. These experimental protocols were approved by the University of Iowa's Institutional Review Board (#201402720).

## Data collection procedure

Participants signed a written informed consent document during a clinic visit prior to surgery. Data collection for this study took place during awake bilateral DBS lead implantation surgery. Before

surgery began, participants practiced the behavioral task. During surgery, two recording sessions took place. Following placement of bilateral subgaleal 4-contact electrode strips (Ad-Tech, Inc) directed posteriorly from the burr holes at the coronal suture so as to sit over SMC, a short recording session (a functional localizer) was used to confirm correct placement of the strip electrodes. Participants performed a short, 40-trial version of the SST that did not include any stop-signals (i.e. it was purely a two-alternative forced-choice reaction time task). These data were analyzed immediately in the operating room and the electrode lead placement was changed if the initial placement did not reveal the typical signature of SMC activity during movement execution (described subsequently in the 'Analyzing local field potentials' section). Then, after the DBS leads (3387, Medtronic, Inc, Minneapolis, MN) were successfully implanted into the bilateral subcortical sites (STN or VIM) using framed indirect stereotactic targeting refined by standard confirmatory physiologic testing (*Gross et al., 2006*; *Geraedts et al., 2019*; *Malinova et al., 2020*). STN localization was confirmed with multi-electrode simultaneous microelectrode recordings to define the dorsal and ventral borders following by multiple sessions of macrostimulation testing of efficacy as well as side effects. VIM localization was confirmed with macrostimulation to achieve tremor reduction efficacy as well as transient contralateral hand paresthesias. Routine post-operative brain imaging was not performed for these participants, but estimated exemplars of recording locations are shown in *Figure 3—figure supplement 4*. Following successful target localization and lead placements, a second recording session took place, which contained the main experiment (see next section).

## Behavioral paradigm

Participants completed an auditory SST (see *Figure 3—figure supplement 1*) in the operating room while recordings were collected. Task stimuli were played through in-ear headphones (ER4 SR model with ER38-14F foam buds, Etymotic Research, Elk Grove Village, IL, USA) connected to a Dell laptop running Fedora, using the PsychToolbox package (version 3; *Brainard, 1997*) in MATLAB (MathWorks, Natick, MA). Participants responded using two USB response buttons held in the hands (Kinesis Savant Elite 2, Kinesis, Bothell, WA). Participants heard a 100ms long, 500 Hz sine wave tone cuing a response (the go signal) every 4 s. Half of the go signals were presented in each ear (in random order); participants were instructed to respond with the button that indicated the side to which the tone was presented. (If the tone was presented in the left ear, the participants pressed the left button, and vice versa.) Participants had 2 s to respond to the tone, after which the task proceeded to a 2 s inter-trial interval.

On one-third of trials, participants heard a second, 1500 Hz tone (the stop signal) presented in both ears, cuing patients to try to stop their response. The delay between the go and stop signal, the stop-signal delay (SSD), was adjusted throughout the task to ideally converge on a stopping accuracy of 50%. Initial SSD was set to 250ms and adjusted in 50ms increments for each hand – subtracting 50ms following failed stops and adding 50ms following successful stops. To prevent proactive strategies, participants were instructed that it was equally important to (1) respond as fast as possible and (2) try to cancel movements successfully when the stop-signal occurred. The pre-surgical practice with the experiment consisted of one block of 30 trials (10 stop). The main task and recording block following macroelectrode lead placement included four blocks of 48 trials (16 stop). Between each block of the main task, the participants rested as needed and received feedback on their performance if necessary.

## Local field potential recordings

Local field potentials (LFPs) were recorded from the thalamus or STN using the four macroelectrode contacts on each DBS lead and from two four-, six-, or eight-contact strip electrodes placed in the subgaleal space over SMC (Ad-Tech, Oak Creek, WI; 10 mm spacing center-to-center, 3 mm exposed contact diameter).

The neurosurgeon (JDWG) inserted the strip electrodes into the subgaleal space posterior to the stereotactic burr hole at the coronal suture, para-sagitally in direction and anterior-posterior in alignment to cover the precentral gyrus. Estimations of the most posterior electrode were ~6 cm posterior to the coronal suture, which is consistent with a posterior placement covering precentral gyrus and SMC (*Park et al., 2007*; *Rivet et al., 2004*). We used the same electrode placement procedure for an identical recording set-up described in *Wessel et al., 2019*. LFP recordings were made on a Tucker-Davis technologies (Alachua, FL) system, using a RA16PA 16-Channel Medusa pre-amplifier

and a RA16LI head-stage. The sampling rate for recording was 24 Hz or 2 Hz, with a low-pass filter of 7.5 kHz on the hardware side. Stimulus onsets were marked in the recording using a TTL pulse from a USB Data Acquisition Device (USB-1208FS, Measurement Computing, Norton, MA) triggered by the stimulus presentation laptop.

## Preprocessing local field potentials

Preprocessing and analysis of LFP data were conducted using custom MATLAB scripts. Data and analysis code for this study can be found on Dryad at https://datadryad.org/stash/dataset/doi:10.5061/dryad.gf1vhhmq0 (*Diesburg et al., 2021*). Electrical line noise from the operating room environment was filtered from the data using EEGLAB's (*Delorme and Makeig, 2004*) *cleanline* function after which the recordings were down-sampled to 1000 Hz for analysis. Then, the recordings were visually inspected for any artifacts. Any 1 s segment of the recording containing an artifact was removed from the data.

## Analyzing local field potentials

Electrode pairs were converted to bipolar montages, resulting in three bipolar recordings from each side of the subcortical location. We conducted LFP analyses using ventral-most (contacts 0 and 1) and dorsal-most (contacts 2 and 3) bipolar arrays in thalamus and the bipolar array in the STN which included the greatest number of β bursts over the entire recording. We utilized the Medtronic lead labeling nomenclature such that contact 0 was the distal-most contact and positioned at the ventral border of each nucleus and contact three was the most proximal contact. Intercontact spacing was 1.5 mm. For thalamic DBS leads, placement of contacts 0 and 1 in the VIM were confirmed with clinical stimulation and testing in the operating room. Specifically, upper extremity tremor reduction was visible and low voltage (i.e. <2 V) paresthesias were achieved in the contralateral hand in all participants; these paresthesias were transient with test stimulation intensities up to 5 V. It is recognized that DBS electrodes placed via coronal / pre-coronal entry points typically span the border of VIM and Vop (*Krack et al., 2002*), making it likely based on these trajectories that LFP recordings from macroelectrodes capture both thalamic regions. Moreover, it is also likely that even recording electrodes that were not on the border between nuclei were nonetheless recording activity from both due to volume conduction.

Broadband event-related spectral perturbation (ERSP) plots of go-locked activity were made using a window of 100ms before stimulus onset to 1500ms following stimulus onset. A baseline window of 500ms to 200ms before stimulus onset was used to perform baseline corrections. Data were converted to time-frequency series using the filter-hilbert method: a Hilbert transform was applied to data filtered at specific frequencies (1–50 Hz) with a window of 0.5 Hz below and above that frequency using symmetric 2-way least-squares finite impulse response filters. The analytic signal was extracted by computing the squared absolute value of the complex signal.

Following localization, go-signal-locked ERSP plots were created for each bipolar array on the two subgaleal strips and visually inspected. One of the researchers in the OR (DAD) visually checked these ERSP plots for a visible, circumscribed decrease in average β band amplitude, a signature of movement-related activity in SMC (such as was observed by *Pfurtscheller and Lopes da Silva, 1999*). If no β suppression was observed in any bipolar array on one or both strips, those strip electrodes were replaced for more optimal positioning. Repositioning of one strip electrode was required in two of the 21 participants and repositioning of both was required in one participant.

## β *burst quantification*

β burst detection was performed using the same procedure as in *Wessel, 2020* and *Shin et al., 2017*. Data from each bipolar electrode array were convolved with a complex Morlet wavelet constructed using the following equation:

$$w\left(t,f\right) = A \, exp\left(-\frac{t^2}{2\sigma_t^2}\right) \, exp\left(2i\pi \, ft\right),$$

with $\sigma = \frac{m}{2\pi f}$, $A = \frac{1}{\sigma_t} \sqrt{2\pi}$, and m = 7 (cycles) for each frequency in the β band (15–29 Hz). This β range was chosen based on foundational studies of cortical β bursts during movement (*Sherman et al., 2016*; *Shin et al., 2017*). The absolute value of the resulting complex data was squared to yield

time-frequency power estimates. The resulting time-frequency data were epoched around events of interest (go and stop signals) with a window of 500ms before stimulus onset to 1000ms after stimulus onset. β bursts were classified by identifying local maxima in the trial-by-trial time-frequency data that exceeded six times the median of the time-frequency power for that specific array across the recording and that lasted at least two β cycles. In other words, the timing of a β burst was quantified only once at its center, at the time of maximal power within its frequency band. β burst frequency was defined as the frequency value at which maximum amplitude was quantified. β burst duration was the time in milliseconds during the search window where the amplitude at burst frequency exceeded the 6xmedian threshold cutoff.

### Lagged coherence

We quantified lagged phase coherence using the approach described in *Fransen et al., 2015*. Analyses were conducted with the FieldTrip software package (*Oostenveld et al., 2011*) using four cycles and frequencies between 8 and 35 Hz (i.e. the β band and surrounding frequencies). Data used in the computation included epoched data from both go and stop trials at all recorded locations.

### Statistical analysis

#### Behavioral analysis

Two participants' data were excluded from analysis because they performed below chance accuracy (50%) on go trials or did not perform the task correctly during the main task. Participants included in the final analysis were 9 STN DBS patients, 11 thalamic DBS patients, and 1 participant with both thalamic and STN DBS. With the exception of the two key analyses that made use of the simultaneous recording of STN and thalamus (i.e. quantifying bursts in subcortical regions between SSD and SSRT, and comparing latencies of bursts in STN and thalamus in the single-subject analysis), the patient with both regions recorded was only included in one of the sample groups – in other words, the participant with data from both STN and thalamus only contributed data to the STN group for most statistical comparisons. Individual task blocks within participants were excluded from analysis if mean accuracy on go trials during a block was less than 60%, or if participants did not successfully stop on at least one stop trial. Based on these criteria, six of the 21 participants had one block of four excluded from behavioral and LFP analysis. Mean accuracies for stop and go trials were extracted for each subject. Go trials were considered incorrect if participants pressed the wrong button or missed responding before the 2 s deadline. Mean RTs for failed stop and successful go trials were extracted for each subject, and SSRT was calculated using the integration method with go omission replacement (*Verbruggen et al., 2019*). Differences in average accuracy and RT measures between STN and thalamic implant patient groups were tested using two-sample *t*-tests.

#### Temporal progression of β bursts

In analyses of the temporal progression of β bursts, we included data from both hemispheres of subcortical sites. This approach is supported by findings that STN has a bilateral representation during movement execution (*Alegre et al., 2005*; *Devos et al., 2006*). Data from SMC was only included from sites contralateral to the correct trial. We quantified counts of bursts across all trials of the same type, binned by burst latency with respect to stimulus (go or stop signal) onset latency, in bins of 100ms from stimulus onset to 900ms following stimulus onset. For matched go trials, β burst latency was calculated with respect to the SSD set in the staircase for that trial. We also quantified bursts during a pre-stop signal baseline to ensure that there were no differences between burst rates across conditions before the stop-signal by summing bursts in the 100ms before SSD and averaging by the total number of trials.

For the analysis in which bursts were time-locked to β bursts at another recording site, the analysis was constrained to bursts within 500ms following the stop-signal (or 500ms following SSD for matched-go trials) in order to assess bursts that would reasonably contribute to movement cancellation based on average sample SSRT (474ms; *Table 1*). We calculated the latency difference between the first subcortical burst following the stop signal and all bursts in SMC during the same trial. This analysis was also repeated with the reverse 'directionality', analyzing subcortical burst rates time-locked to SMC β bursts in the same manner.

Permutation-based statistics were used to evaluate statistical significance. Specifically, two-way repeated measures ANOVAs were calculated with factors of *TIMEPOINT* and *TRIAL TYPE*. ANOVAs were bootstrapped by comparing resulting $F$ values for each factor to null distributions of $F$ values from 10,000 tests with data labels randomized. True $F$ values were considered significant if they were greater than the $F$ value at the $95^{th}$ percentile of the null distribution and $p < 0.05$. Pairwise differences between trial types at specific time points were calculated using $t$-tests with Bonferroni-Holm corrections for multiple comparisons.

## Calculating bursts in SSD-SSRT period

To calculate subcortical burst rate differences between failed and successful stops and matched go trials in the SSD-SSRT delay, we quantified the total number of bursts between trial-specific SSD and the participant's average SSRT (in other words, between time of SSD and SSD plus SSRT). Matched go trials were split into fast and slow trials using a median split of participant-wise go RTs. The participant with both thalamic and STN recordings contributed both STN and thalamic recording data to this analysis.

## Timing of β bursts at each subcortical recording site

To delineate the relative timing of bursts at different sites during stop trials, we calculated the mean latency of first bursts with respect to both stop-signal onset and subject-wise SSRT at each recording location following stop-signal onset. These first bursts were quantified between stop-signal onset and one second following stop-signal onset. This analysis was also performed in the subject with STN and thalamic recordings. We conducted a between- and within- subjects ANOVA with a within-subject factor of *TRIAL TYPE* and between-subjects factor of *SUBCORTICAL LOCATION* to assess whether successful stops might be associated with a shorter delay between STN and thalamic bursts than failed stops.

# Acknowledgements

The authors thank Haiming Chen for assistance with surgical recordings and the patient participants for volunteering their time. This research was funded by an NIH fellowship (T32GM108540) to DAD, Carver College of Medicine/Iowa Neuroscience Institute Research Program of Excellence funding to JDWG and JRW, and grants from the NIH (R01NS117753) and NSF (CAREER 1752355) to JRW.

# Additional information

## Funding

| Funder | Grant reference number | Author |
|---|---|---|
| National Institutes of Health | T32GM108540 | Darcy A Diesburg |
| National Institutes of Health | R01NS117753 | Jan R Wessel |
| National Science Foundation | CAREER 1752355 | Jan R Wessel |
| Carver College of Medicine & Iowa Neuroscience Institute | Research Program of Excellence Funding | Jeremy Greenlee Jan R Wessel |

The funders had no role in study design, data collection and interpretation, or the decision to submit the work for publication.

## Author contributions

Darcy A Diesburg, Conceptualization, Formal analysis, Investigation, Methodology, Software, Visualization, Writing – original draft, Writing – review and editing; Jeremy DW Greenlee, Conceptualization, Investigation, Methodology, Supervision, Writing – review and editing; Jan R Wessel,

Conceptualization, Formal analysis, Funding acquisition, Investigation, Methodology, Software, Supervision, Writing – original draft, Writing – review and editing

### Author ORCIDs
Darcy A Diesburg ![ORCID] http://orcid.org/0000-0002-3489-7624
Jeremy DW Greenlee ![ORCID] http://orcid.org/0000-0002-8481-8517
Jan R Wessel ![ORCID] http://orcid.org/0000-0002-7298-6601

### Ethics
Human subjects: Research participants signed a written informed consent document during a clinic visit prior to surgery. Experimental protocols were approved by the University of Iowa's Institutional Review Board (#201402720).

### Decision letter and Author response
Decision letter https://doi.org/10.7554/eLife.70270.sa1
Author response https://doi.org/10.7554/eLife.70270.sa2

## Additional files

### Supplementary files
• Transparent reporting form

• Supplementary file 1. Information about participant diagnoses, handedness, symptom laterality, and pre-operative symptom severity scores. For participants with Parkinson's disease, scores shown are from the motor examination portion (i.e., total part III) of the Unified Parkinson's disease rating scale (UPDRS). For essential tremor participants, scores are from the Fahn-Tolosa tremor scale. All Parkinson's disease patients besides participant 2 were diagnosed with idiopathic Parkinson's disease. Participant 2 had a diagnosis of both idiopathic Parkinson's and essential tremor.

### Data availability
All data analyzed during this study and scripts used for analyses are available on Dryad.

The following dataset was generated:

| Author(s) | Year | Dataset title | Dataset URL | Database and Identifier |
|-----------|------|---------------|-------------|--------------------------|
| Diesburg DA, Greenlee JDW, Wessel JR | 2021 | Cortico-subcortical β burst dynamics underlying movement cancellation in humans | https://doi.org/10.5061/dryad.gf1vhhmq0 | Dryad Digital Repository, 10.5061/dryad.gf1vhhmq0 |

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
