## [Editor Report]

This work makes an important contribution to the literature and addresses timely and interesting questions relating to the role of transient beta oscillations in cancelling motor responses in a rare and valuable dataset.

---

## [Decision Letter]

**Decision letter after peer review:**

Thank you for submitting your article "Cortico-subcortical β burst dynamics underlying movement cancellation in humans" for consideration by *eLife*. Your article has been reviewed by 3 peer reviewers, and the evaluation has been overseen by a Reviewing Editor and Richard Ivry as the Senior Editor. The following individuals involved in review of your submission have agreed to reveal their identity: Vignesh Muralidharan (Reviewer #2); Robert Schmidt (Reviewer #3).

Essential revisions:

In general, the reviewers were impressed with the manuscript – especially the novelty of the data. There were some concerns, especially with respect to the interpretation of results. These could be mitigated by some additional analyses. Below are more detailed comments.

1) A main conclusion is summarized in the 'fronto-basal ganglia network model for inhibitory control' in Figure 4, in which the thalamus sends the basal ganglia beta oscillations back to cortex. While this mechanism is of course possible, it does not seem to be supported by several results of the manuscript. In Figure 3 there is no indication of SMC beta bursts following the thalamus, which does not fit to their model. Although it is noted that the results for dorsal and ventral thalamus are remarkably similar, there seems to be several differences in key analyses. For example the timing of beta bursts and distinctions between trials types (Figure 2, right panels) seem to differ between ventral and dorsal thalamus (and actually precede STN, contradicting the model). Furthermore, only ventral, but not dorsal thalamus seems to have higher beta burst rates when stopping is successful (Figure 2, left). The key analysis of the relative timing using simultaneous recordings (Figure 4B) shows that beta in dorsal, but not ventral thalamus is later than in STN. The same analysis involving comparisons between the patient groups (Figure 4A) does not show any differences between STN and dorsal or ventral thalamus timing for the first burst, but only for the later beta that occurs after the SSRT (Figure 4A, right panel). However, as the model seems to rely on the very fast propagation of beta, it does not seem to be consistent with these results. Finally, the authors show a clear effect of TIMEPOINT in both these scenarios, but the comparison in the plots seem to be between trial types. Why was this was done as in both cases there wasn't a TIMEPOINT x TRIAL TYPE interaction? Shouldn't the comparison be across the different time-points, for instance the one before and the one after STN/thalamic burst to substantiate the claims of increased bursting of SMC after the sub-cortical nuclei?

2) In Figure 4, it is not clear (we might have missed this) what is the temporal window considered for getting the time of the first burst in STN and thalamus? The authors show the average first burst time occur in STN prior to the thalamus, providing support for the STN-thalamus part of the temporal model of stopping. If the first burst indexes recruitment of the inhibitory system via STN, could the authors also show the same average first burst time for the SMC. It would nice to see whether the model shown in Figure 4B extends till the SMC. Although they do this in Figure 3, that is in relation to STN bursts. If this does not pan out discuss why it is not the case, and what they might be capturing.

3) In Figure 1 several analyses are shown about beta decreasing during movement initiation. However, the plots seem to be aligned to the onset of the Go stimulus. Therefore, it is difficult to judge the time relation between the beta decrease and the initiation of the movement. Instead, or in addition, it would be helpful to see the data aligned to movement initiation, with a sufficient pre-event period, in order to see whether beta decreases during movement.

4) The timing of certain processes seemed a bit too late to fit in the temporal model of stopping. Keeping in mind that this is a patient population and the responses overall are slow and the SSRTs are also longer, the emergence of first beta activity in the STN seems to be close to SSRT (~400ms, Figure 4), whereas a robust signature of motor inhibition, the global motor suppression has been observed to happen much earlier (~140-150ms after stop-signal), which indexes the suppression of the motor cortex via inhibition exerted by the STN on thalamus. How do the authors reconcile those observations with their findings?

5) With respect to above point, in Figure 2, most of the changes in the beta burst rate seem to be happening after group average SSRT in both the STN and thalamic group and not prior to it. I appreciate that the authors acknowledge this lack of effect seen in the pre-SSRT burst rates which they suggest could because the individual variability in the SSRT has not been taken into account. I suggest that the authors repeat this analysis but time-lock the bursts to individual's SSRT (similarly in matched go trials) and see if they find out if the relationship comes out more robustly. Also, there is evidence of prefrontal burst times correlating with SSRT in previous studies (Jana et al. 2020) i.e. later bursts are associated with later SSRT. So, the authors could look at if the average time of the STN bursts (in successful stop trials) prior to the individual SSRT correlate with it, trying to explain this variability. Also in Figure 4B for single subject exemplar, could the authors also show the timing of the first burst in the STN and thalamus in relation to the SSRT (or mark it on the Figure), to see if the timing of sub-cortical regions is at least on average prior to it?

6) For many of the described results it is also important to consider how beta bursts relate to reaction times in Go trials. In Figure 1C average reaction times are determined for trials with and without beta oscillations. Depending on how this analysis is done, there is a potential problem due to counting beta bursts in time windows of different lengths. For trials with long reaction times, by definition, the time window between Go cue and movement response is longer than for trials with short reaction times. Therefore, even if beta bursts occur with a fixed probability at any time, it would result in a higher count of beta bursts in long reaction time trials, simply due to a longer time window considered. Thereby comparisons between groups of trials with and without beta bursts would be biased to yield long and short reaction times, respectively. This could be avoided by normalizing the burst count by the length of the time window (ie look at the burst rate instead). In any case, due to the potential relation between beta bursts and reaction time, it would be important to also consider Go trials with different reaction times for comparison with stop trials (Figure 2).

7) The SST has the inherent characteristic of comparing a motor to a non-motor output. When investigating beta activity this is obviously a problem if we want to discern "status quo" (Engel and Fries, 2010) from "inhibitory" signaling. In addition to comparing beta bursting from Go/Failed stop trials (movement) with successful strop trials (no movement), the authors could consider to compare beta burst rate during successful stopping with "rest" activity (here f.ex. during the inter-trial interval): Does beta bursting increase during stopping when compared to the "status quo" activity at rest? We note that the authors did perform compare results to pre-stop burst rates, but we were unclear if this baseline was immediately prior to the stop signal or during a resting period when no movement would be expected.

8) It would be intriguing to compare beta burst properties across SMC, thalamus, and STN, such beta burst wave shape and burst duration. Furthermore, it is referred to previous studies that had supported that beta seems to primarily occur in brief bursts (rather than slow, continuous power modulations). Due to the unique nature of this data set with simultaneous recordings from multiple sites in humans, it would be valuable to establish and verify these assumptions here, as the analyses done here seem to build on them.

9) In the Results the SSRT differences between groups seem quite large (>100ms), so it would be helpful to include the details on the statistical tests done to compare SSRT between thalamus and STN patients (even if differences are not statistically significant).

10) Although the main motivation of the paper is to look at beta bursts, it would be good if the authors also showed typical ERSP results from the STN and thalamus for the stop and go trials. Specifically, is there a relative change in beta power between the successful stop and matched go trials in the period between the stop-signal and SSRT, the difference ERSP will help. This might be confirmatory to the observation of the increased beta bursts seen during the successful stop.

11) Is there a lateralization of the findings? The literature often emphasizes the inhibitory network to be right hemispheric. It seems the authors averaged across hemispheres – it might be interesting to investigate whether there is a differential contribution of right and left, if possible.

12) To show that reported effects are frequency-specific, It could be interesting to add a control frequency (for instance. low frequency or gamma activity).

13) In the manuscript it is mentioned that response inhibition here is about cancelling already-initiated movements (e.g. "When already-initiated movements had to be cancelled…" in the impact statement). However, this does not seem to fit to the stop-signal task, especially with rather ballistic actions such as a button press. We assume that in most cases the action is stopped before any movement starts, and not e.g. while the finger is in the process of moving to make the button press (as that time window is very short for ballistic actions). Please clarify in the manuscript. Similarly, there are several statements about beta reflecting a "rapid re-instantiation of motor inhibition". It was not clear to me how this is concluded what it actually means (why 're'-instantiate?). Furthermore, it is stated that the findings have 'strong implications for many movement disorders' (eg in the Impact Statement). What are these implications, can they be spelled out? Similarly, in the Discussion it is stated that "this has major implications for emerging theories of movement cancellation". What are these implications?

14) In the explanation at the bottom of page 13, it is argued that the participant-specific SSRT accounts for the differences between analyses. Could the difference also simply be explained by one analysis involving (arbitrary) binning, while the other integrates differences across several bins into a single time window?

*Reviewer #1:*

In this study, the authors investigate the association between beta bursting and inhibitory processing in several motor circuit nodes (SMC with STN or VIM) of 21 patients with either PD or ET. They show that subcortical beta bursting is increased when stopping of initiated movements is successful. During successful stopping, specifically subthalamic beta bursts are preceding those in SMC. The authors claim that these results provide evidence for the role of beta bursts in conveying inhibitory signaling through the CBGTC-loop as well as for the directionality of this signaling (STN → VIM → SMC).

The paper is well written, results are presented in a clear way, figures are showing individual data points in comprehensive box-plots and the discussion is extensive and openly addresses limitations. The research question of an association between beta bursting and inhibitory processes as well as its directionality is intuitive and important. The presented data is valuable per se, as parallel cortico-subcortical intracranial recordings during task-performance in humans are rare.

While the authors address the limitations openly, I am still not entirely convinced that some of the claims are sufficiently backed up by the acquired data and current state of the literature. First, I am concerned by mixing different disease entities (PD and ET). I am not an expert in the field, but I think that there is an ongoing debate whether the dopamine-depleted state of PD is associated with alterations in response inhibition (for ex. Manza et al., 2017), which makes it difficult to present the PD-OFF state as "physiological". Second, it is unfortunate that there apparently has been no post-operative imaging in this cohort, as this would significantly help to undermine the claim of dorsal contact pairs capturing Vop instead of VIM activity. Third, I got the impression that the finding of directionality was slightly over-emphasized, given that thalamic bursting was not shown to precede SMC and that this would be a relevant pre-requisite for the hypothesis that subthalamic beta activity is conveyed through the thalamus to SMC.

*Reviewer #2:*

The paper explores the cortical-subcortical (basal-ganglia-thalamus- sensorimotor cortex) dynamics during human action-stopping, looking at transient beta (15-29Hz) events/bursts during stopping. The authors' findings support, especially at the sub-cortical level, the temporal model of stopping which implicates a frontal-basal-ganglia-thalamocortical circuitry during rapid action-stopping. The paper shows that beta bursts, a potential signature of inhibitory information flowing through this circuitry, occur earlier in the STN followed by the motor thalamus and sensorimotor cortex, fitting with the idea that STN is recruited first to inhibit the thalamus and thus movement. By recording simultaneous LFP from key nodes in the action-stopping network, the authors have high spatiotemporal resolution to investigate the dynamics in these regions which is a highlight of this study. Furthermore, by looking at the transient beta events in these regions they are able to estimate the exact time of beta activity and show timing specificity of sub-cortical nuclei during stopping.

Although the paper presents a good picture of the temporal model, there seems to be few caveats specifically regarding the beta burst time course, for instance most of the robust changes in the burst activity across trials seems to be happening after average stopping time (aka stop-signal reaction time) in both the STN and thalamus. In addition, the timing of first STN and thalamic beta burst also seems to be happening on an average close to and after SSRT respectively (~400-500ms). This seems a bit too late in respect to other metrics of motor inhibition, for instance the STN mediated motor suppression which seems to occur much earlier (~140-150ms in relation to a signal to stop). However, the authors show increased beta bursting in the sensorimotor cortex after STN beta bursts and higher number of bursts in sensorimotor cortex leading to slower reaction times, which is good demonstration of the potential role of the basal-ganglia-thalamocortical circuitry in aiding action stopping.

*Reviewer #3:*

In their manuscript "Cortico-subcortical β burst dynamics underlying movement cancellation in humans", Diesburg et al. examine beta oscillations in humans performing a stop-signal task. The beta oscillations are recorded in sensorimotor cortex (SMC) and either thalamus or subthalamic nucleus (STN; one patient has recordings in both subcortical sites) in patients with Parkinson's disease and Essential Tremor. The focus is on the relative timing of beta bursts with the suggestion that they propagate from STN over thalamus to SMC to mediate inhibitory control. Based on several analyses, it is stated that beta bursts increase when stopping is successful, and that STN beta bursts precede SMC beta bursts by up to 50ms. Furthermore some evidence is shown that STN beta also precedes thalamus. It is concluded that the recordings in humans support network models in which beta is quickly evoked by stop-signals and then propagated through cortical and subcortical loops.

A major strength of this paper is that it provides an important connection between previous studies with invasive recordings in non-human animals and human studies with non-invasive recordings. Thereby the manuscript achieves to connect findings about beta oscillations in the basal ganglia, thalamus and cortex. A (related) weakness of the paper is that the size of the data set is constrained due to the nature of the recordings (e.g. only one patient with simultaneous STN and thalamus recordings), and that some of the results rely on comparisons between different patient groups (with potentially different beta burst dynamics). These limitations are considered and addressed well by the authors by providing a comprehensive set of analyses. However, in some cases it is not clear whether the analyses and results support the statements and claims made.

---

## [Author Response]

Essential revisions:In general, the reviewers were impressed with the manuscript – especially the novelty of the data. There were some concerns, especially with respect to the interpretation of results. These could be mitigated by some additional analyses. Below are more detailed comments.1) A main conclusion is summarized in the 'fronto-basal ganglia network model for inhibitory control' in Figure 4, in which the thalamus sends the basal ganglia beta oscillations back to cortex. While this mechanism is of course possible, it does not seem to be supported by several results of the manuscript. In Figure 3 there is no indication of SMC beta bursts following the thalamus, which does not fit to their model. Although it is noted that the results for dorsal and ventral thalamus are remarkably similar, there seems to be several differences in key analyses.For example the timing of beta bursts and distinctions between trials types (Figure 2, right panels) seem to differ between ventral and dorsal thalamus (and actually precede STN, contradicting the model). Furthermore, only ventral, but not dorsal thalamus seems to have higher beta burst rates when stopping is successful (Figure 2, left). The key analysis of the relative timing using simultaneous recordings (Figure 4B) shows that beta in dorsal, but not ventral thalamus is later than in STN. The same analysis involving comparisons between the patient groups (Figure 4A) does not show any differences between STN and dorsal or ventral thalamus timing for the first burst, but only for the later beta that occurs after the SSRT (Figure 4A, right panel). However, as the model seems to rely on the very fast propagation of beta, it does not seem to be consistent with these results. Finally, the authors show a clear effect of TIMEPOINT in both these scenarios, but the comparison in the plots seem to be between trial types. Why was this was done as in both cases there wasn't a TIMEPOINT x TRIAL TYPE interaction? Shouldn't the comparison be across the different time-points, for instance the one before and the one after STN/thalamic burst to substantiate the claims of increased bursting of SMC after the sub-cortical nuclei?

The reviewers raise an excellent point. Indeed, the lack of SMC bursts following thalamic bursts was at first surprising to us as well. There is, of course, the possibility that the somewhat arbitrary time bins for this analysis obscure a potential effect. However, we do also have to take seriously the possibility of a true absence of such beta dynamics between Th and SMC. Indeed, while we show that rapid motor inhibition is signified by beta burst activity in STN and SMC (moreover in a sensible temporal sequence), it is very possible that Th-SMC communication does not take place via burst like beta activity. We have added a paragraph to our Discussion (on p. 28) expanding on this possibility.

“While our results support the view that beta bursts signify motor inhibition on the level of the basal ganglia, thalamus, and SMC, we did not find the same systematic temporal relationship between beta bursts in thalamus and in SMC that was found for STN and SMC – i.e., the ostensible subcortical start- and cortical endpoints of the cascade. et al.[…] In this scenario, the tight temporal relationship between the emergence of beta in STN and SMC during stopping would be merely an indirect effect of the fact that beta bursts signify the rapid re-instantiation of inhibition at both levels, rather than directly reflecting the propagation of beta commands through the entire basal ganglia-thalamus-SMC chain.”

Beyond this point, however, there are also a few misunderstandings about the results. These are (hopefully) addressed below and via changes to the revised manuscript. In detail, these are as follows:

“For example the timing of beta bursts and distinctions between trials types (Figure 2, right panels) seem to differ between ventral and dorsal thalamus (and actually precede STN, contradicting the model).”

It is true that the dorsal thalamus recordings in our sample did not contain differences in burst counts in the SSD-to-SSRT window. However, Figure 2 (now Figure 3) does not contain any information about relative timing between STN and thalamic bursts across the two groups. We suspect the reviewer is referring to the gray lines marked on the right panel, which simply show the average SSRT for STN and thalamic DBS patients, and do not indicate a particular time point at which beta bursts were plotted (they were counted between trial-wise SSD and individual subject SSRT). The only depiction of burst timing with respect to individual SSRT is shown in Figure 4 (now Figure 5), wherein thalamic bursts follow STN bursts on average. Please note that this is especially remarkable given that SSRTs were longer for our STN group on average than the thalamic group. We have edited the manuscript at several points on pages 17, 18, 23, and 24 to hopefully make this clearer.

“The same analysis involving comparisons between the patient groups (Figure 4A) does not show any differences between STN and dorsal or ventral thalamus timing for the first burst, but only for the later beta that occurs after the SSRT”.

Both panels in 4A (now Figure 5A) show the timing of the same beta bursts (the first bursts in a respective region during the SSD-SSRT period), but with different time locking. The left plot shows the first burst with respect to the stop-signal and the right one shows the first burst with respect to SSRT. The fact that burst timing significantly differs when the data are SSRT-locked speaks in favor of the proposition that stop-related beta bursts seem tightly linked to individual SSRT (and therefore the timing of each individual’s stopping process).

“Finally, the authors show a clear effect of TIMEPOINT in both these scenarios, but the comparison in the plots seem to be between trial types”

The reviewers are correct that Figures 2A and 3A (now Figures 3 and 4) do not display TIMEPOINT effects, and the significant comparisons in Figure 3A are effects of TRIAL TYPE on burst counts (panel A) or on burst rates (C). In Figure 3, after we found main effects of both TIMEPOINT and TRIAL TYPE for stimulus-locked burst rates, we used follow-up t-tests to determine at which time points comparisons between successful stop trials and other trials were significant. The reviewer is also correct that we only found a main effect of TIMEPOINT in the burst-locked analysis (shown in Figure 4), and no main effect of TRIAL TYPE or interaction effect. In that instance, we chose to conduct pairwise tests between trial types anyway, specifically because of our strong a priori hypothesis. Indeed, this analysis is essentially a repetition of the exact analysis from the Wessel 2020 J Neuro paper, except that the scalp-site FCz is replaced with the STN LFP. We’ve highlighted this decision in the text now on p. 19

We did not test for a TIMEPOINT effect in the analyses depicted in Figure 4 (now 5) – those analyses tested for effects of TRIAL TYPE and BURST LOCATION. We have added detail to all our Figure captions to be clearer about which comparisons are plotted. In panel C, the pairwise comparisons are follow-up tests comparing trial types at specific time bins, following up on significant omnibus effects of TRIAL TYPE (described on p. 14 of the Results section). In Figure 5, the significant comparisons shown in the Figure were effects of BURST LOCATION on average timing of first bursts following the stop signal.

We apologize for these misunderstandings and hope that these revisions have made things clearer.

2) In Figure 4, it is not clear (we might have missed this) what is the temporal window considered for getting the time of the first burst in STN and thalamus? The authors show the average first burst time occur in STN prior to the thalamus, providing support for the STN-thalamus part of the temporal model of stopping. If the first burst indexes recruitment of the inhibitory system via STN, could the authors also show the same average first burst time for the SMC. It would nice to see whether the model shown in Figure 4B extends till the SMC. Although they do this in Figure 3, that is in relation to STN bursts. If this does not pan out discuss why it is not the case, and what they might be capturing.

The window for obtaining the first bursts was one second from the stop-signal onset (or from SSD for go trials). We have added this detail to the figure caption and the Methods section.

In terms of the first-burst timing in SMC, we did not include those data in the figure on purpose. That’s because SMC produces a steady bursting of beta at baseline (Kilavik et al., 2013; Little et al., 2019). Importantly, this burst rate is upregulated when proactive control is exerted, such as in the stop-signal task (Soh et al., 2021). These proactive control-related bursts are impossible to distinguish from any event-related increases incurred by the stop-signal (which we were interested in in this study) – unless the data are plotted directly time-locked to beta bursts in other regions (c.f., Wessel 2020 J Neuro and Figure 4). Hence, it is impossible to know if an SMC burst was truly triggered by the stop-signal, or merely a reflection of a state of heightened proactive control. Instead, we believe that the analysis in which SMC activity is time-locked to bursts in subcortical areas (cf., Figure 4) is the best way to investigate SMC to test our hypothesis about subcortical-cortical communication. We hope the reviewers agree.

3) In Figure 1 several analyses are shown about beta decreasing during movement initiation. However, the plots seem to be aligned to the onset of the Go stimulus. Therefore, it is difficult to judge the time relation between the beta decrease and the initiation of the movement. Instead, or in addition, it would be helpful to see the data aligned to movement initiation, with a sufficient pre-event period, in order to see whether beta decreases during movement.

We agree. We have added a panel to Figure 1 including beta burst rates at individual time bins locked to responses.

4) The timing of certain processes seemed a bit too late to fit in the temporal model of stopping. Keeping in mind that this is a patient population and the responses overall are slow and the SSRTs are also longer, the emergence of first beta activity in the STN seems to be close to SSRT (~400ms, Figure 4), whereas a robust signature of motor inhibition, the global motor suppression has been observed to happen much earlier (~140-150ms after stop-signal), which indexes the suppression of the motor cortex via inhibition exerted by the STN on thalamus. How do the authors reconcile those observations with their findings?

Indeed, in healthy young participants, global motor suppression (indexed by MEPs or EMG) has been shown to occur around 140-150ms following the stop signal. However, the same has not been demonstrated in movement disorder patients, who have much longer SSRTs. Indeed, Jana et al. (2020) observed that SSRT measured using the classic integration method overestimated stopping measured in EMG traces by approximately 60ms. Therefore, we would argue that global motor suppression in our patient sample would be expected to occur approximately 60ms before SSRT as well, and not 150ms after the stop-signal. While the Stop+150ms timepoints and the SSRT-60ms timepoints are very close to one another in healthy humans, in our current population, they are much further apart.

However, it is important to note that STN bursts on average across the sample still do emerge before SSRT, which is an indication to us that the mechanism they index occurs in time to contribute to reactive cancellation, and perhaps even aligns with the SSRT-60ms timing that has been proposed based on healthy adults (cf., the new panel in Figure 3). Of course, in the absence of EMG measures, that is merely speculation.

5) With respect to above point, in Figure 2, most of the changes in the beta burst rate seem to be happening after group average SSRT in both the STN and thalamic group and not prior to it. I appreciate that the authors acknowledge this lack of effect seen in the pre-SSRT burst rates which they suggest could because the individual variability in the SSRT has not been taken into account. I suggest that the authors repeat this analysis but time-lock the bursts to individual's SSRT (similarly in matched go trials) and see if they find out if the relationship comes out more robustly. Also, there is evidence of prefrontal burst times correlating with SSRT in previous studies (Jana et al. 2020) i.e. later bursts are associated with later SSRT. So, the authors could look at if the average time of the STN bursts (in successful stop trials) prior to the individual SSRT correlate with it, trying to explain this variability. Also in Figure 4B for single subject exemplar, could the authors also show the timing of the first burst in the STN and thalamus in relation to the SSRT (or mark it on the Figure), to see if the timing of sub-cortical regions is at least on average prior to it?

Indeed, we believe that trial-type-wise differences in burst rates might not emerge at individual time bins because the true condition differences are tightly linked to subject-wise SSRT. We added plots of individual time bins locked to participant-wise SSRT to Figure 2 (now Figure 3) in the revised manuscript. Increased beta burst rates in the bin immediately preceding SSRT are indeed visible on this new figure, with significant differences seen between STN burst rates for successful stop and go trials in the 100ms preceding SSRT.

Furthermore, while we agree that cross subject correlations between behavior and beta bursts could be informative, we do not believe that our sample size is sufficient to validly allow for such analyses.

Lastly, we have added the single subject’s SSRT (304ms) to Figure 5. Because unlike for the full-sample analysis, all analyses here were performed on the same subject with the same SSRT, the stop- and SSRT-locked analyses and plots are identical.

6) For many of the described results it is also important to consider how beta bursts relate to reaction times in Go trials. In Figure 1C average reaction times are determined for trials with and without beta oscillations. Depending on how this analysis is done, there is a potential problem due to counting beta bursts in time windows of different lengths. For trials with long reaction times, by definition, the time window between Go cue and movement response is longer than for trials with short reaction times. Therefore, even if beta bursts occur with a fixed probability at any time, it would result in a higher count of beta bursts in long reaction time trials, simply due to a longer time window considered. Thereby comparisons between groups of trials with and without beta bursts would be biased to yield long and short reaction times, respectively. This could be avoided by normalizing the burst count by the length of the time window (ie look at the burst rate instead). In any case, due to the potential relation between beta bursts and reaction time, it would be important to also consider Go trials with different reaction times for comparison with stop trials (Figure 2).

First, we agree with the reviewers that there was indeed a systematic difference in search window size between these two trial types, which biased that analysis. Thank you for catching that aspect we had missed! Unfortunately, normalizing burst counts does not fully address this problem either, as time windows with zero bursts are more likely to occur for shorter detection windows / faster RT trials, but wouldn’t be susceptible to normalization (as 0 divided by anything is still 0). To try to address this concern somewhat, we repeated this analysis using a fixed time window of 500ms following the go signal (in all trials with RT > 500ms). While the results of this analysis are numerically in line with what Little and colleagues observed (increased RT in burst containing trials), they were not significant in our sample (though this is due to one participant who strongly deviates from the rest of sample, see (Author response image 1) ). Because these revised results are therefore inconclusive (neither indicative of a failed replication of the Little study, nor actually replicating it), we’ve removed them from the manuscript.

**Author response image 1. sa2fig1:** Main effect of trial type (go + fail): F = 35.78, p <.0001; no main effect of burst presence: F = 0.13, p = .72; no interaction effect: F = 0.88, p = .36.

On the other point, we definitely agree that it is important to consider the different GoRT distribution when comparing subcortical bursts between successful/failed stop vs. go trials. Therefore, we have separated the matched go trials presented in Figure 3B into slow and fast go trials based on a median RT split, as is typical for this approach. Fast Go trials were then compared to failed stop trials and slow Go trials to successful stop trials. The main effects of trial type that were previously reported remained significant. At all recording locations, successful stop trials contain significantly more bursts in the SSD-SSRT window compared to slow go trials, and more bursts than failed stops in the STN and ventral thalamus. These results have been updated in Figure 3 and in the text.

7) The SST has the inherent characteristic of comparing a motor to a non-motor output. When investigating beta activity this is obviously a problem if we want to discern "status quo" (Engel and Fries, 2010) from "inhibitory" signaling. In addition to comparing beta bursting from Go/Failed stop trials (movement) with successful strop trials (no movement), the authors could consider to compare beta burst rate during successful stopping with "rest" activity (here f.ex. during the inter-trial interval): Does beta bursting increase during stopping when compared to the "status quo" activity at rest? We note that the authors did perform compare results to pre-stop burst rates, but we were unclear if this baseline was immediately prior to the stop signal or during a resting period when no movement would be expected.

We disagree with the argument that inhibitory signaling in purportedly inhibitory brain regions like the basal ganglia or M1 has to exceed baseline activity to signify inhibition. The baseline ‘status quo’ in the motoric regions of the basal ganglia and M1 in the absence of movement is inhibition (Uno and Yoshida, 1975; Di Chiara et al., 1979; Yoshida and Omata, 1979; DeLong and Georgopoulos, 1981). While it may be true that activity in control-related prefrontal regions that are not signaling inhibitory commands at baseline would have to show an above baseline increase, the same logic, to us, does not apply to the BG or M1.

Further, the addition of a pre-go-signal baseline (which has been added to Figure 3 and associated in-text results) supports this account. In line with our hypothesis that increasing beta burst rates during stopping reflect a re-instantiation of tonic inhibition present at rest, we see increased beta bursts in early post-stop-signal time bins compared to a pre-stop baseline (because beta bursts decrease during movement execution, signifying net-disinhibition), but no differences in burst rates when comparing to a pre-go baseline (when we assume tonic inhibition is being exerted). We have added these results on p. 16 of the Results section.

8) It would be intriguing to compare beta burst properties across SMC, thalamus, and STN, such beta burst wave shape and burst duration. Furthermore, it is referred to previous studies that had supported that beta seems to primarily occur in brief bursts (rather than slow, continuous power modulations). Due to the unique nature of this data set with simultaneous recordings from multiple sites in humans, it would be valuable to establish and verify these assumptions here, as the analyses done here seem to build on them.

In response to this valuable suggestion, we have extracted average beta burst duration and frequency of peak power for each recorded region. These values have been added to the Results section on p. 10 and a description of this analysis added to the Methods on p. 37.

To confirm the assumption that beta is indeed burst-like, we conducted a lagged coherence analysis. Lagged coherence (Fransen et al., 2015) provides an estimate of how much the phase of a signal predicts future phase at the same frequency. Signals that are oscillatory can be expected to predict their own phase many cycles in the future, while transient signals will have low lagged coherence several phases on. This is indeed what we observe in the 15-29Hz range at all our recording locations: lagged coherence decreases from 1-3 cycles, and at 3 cycles a trough in coherence is observed in the beta band compared to surrounding frequencies. This coherence trough is most pronounced in the SMC, while STN contains activity that is somewhat less “bursty” based on lagged coherence (we note in the text it is possible this could reflect the presence of pathologically long STN bursts in our patient sample). Beta activity in the thalamus seems to be the least burst-like and most sustained of all recording regions. We have added a new Figure (now Figure 2) with these results, an explanation of this approach in the Results section on p. 10, and a description of the methodology on page 38.

9) In the Results the SSRT differences between groups seem quite large (>100ms), so it would be helpful to include the details on the statistical tests done to compare SSRT between thalamus and STN patients (even if differences are not statistically significant).

We agree. The statistics for the other behavioral metric comparisons across groups have been added to the Results section on p. 7.

10) Although the main motivation of the paper is to look at beta bursts, it would be good if the authors also showed typical ERSP results from the STN and thalamus for the stop and go trials. Specifically, is there a relative change in beta power between the successful stop and matched go trials in the period between the stop-signal and SSRT, the difference ERSP will help. This might be confirmatory to the observation of the increased beta bursts seen during the successful stop.

Agreed. We have added ERSPs from STN and thalamus to Figure 2 (now Figure 3), plotting the difference in beta power between successful stop and go trials. There are clear increases in beta band power in STN and thalamus following the stop signal during stop trials.

11) Is there a lateralization of the findings? The literature often emphasizes the inhibitory network to be right hemispheric. It seems the authors averaged across hemispheres – it might be interesting to investigate whether there is a differential contribution of right and left, if possible.

Though the ventrolateral PFC aspects of the fronto-basal ganglia network (i.e., rIFC) are right-lateralized, we are not aware of intracranial research suggesting that the subcortical aspects are. Indeed, STN activity is bilateral during movement execution (Alegre et al., 2005; Devos et al., 2006). While we are aware of one fMRI study that suggests a lateralization on the level of STN (Aron and Poldrack, J Neuro 2006), that study was conducted at 3T, which does not have sufficient signal to noise ratio to reliably capture STN activity (de Hollander et al., Human Brain Mapping, 2017; Miletic et al., NeuroImage, 2020).

Therefore, we chose to pool subcortical bursts across hemispheres in the main manuscript. However, we have added the results separated by hemisphere in Supplement 3 to Figure 3.

12) To show that reported effects are frequency-specific, It could be interesting to add a control frequency (for instance. low frequency or gamma activity).

This is an interesting suggestion. Though gamma is probably burst-like, to our knowledge, however, it has not been shown that low frequencies are similarly burst-like compared to beta. Furthermore, the dominant models which informed our a priori hypothesis pertained specifically to beta. If the reviewers can point us to literature that has assessed bursts during movement execution or cancellation in other frequencies (so that a pre-existing approach can be adopted), we are willing to consider trying this.

However, based on our lagged coherence analysis (referenced above in response 8), there appears to be a peak of “burstiness” in the beta band that compared to the surrounding frequencies. We believe that the included lagged coherence analysis will help readers to compare “burstiness” of beta to neighboring frequency bands.

13) In the manuscript it is mentioned that response inhibition here is about cancelling already-initiated movements (e.g. "When already-initiated movements had to be cancelled…" in the impact statement). However, this does not seem to fit to the stop-signal task, especially with rather ballistic actions such as a button press. We assume that in most cases the action is stopped before any movement starts, and not e.g. while the finger is in the process of moving to make the button press (as that time window is very short for ballistic actions). Please clarify in the manuscript. Similarly, there are several statements about beta reflecting a "rapid re-instantiation of motor inhibition". It was not clear to me how this is concluded what it actually means (why 're'-instantiate?). Furthermore, it is stated that the findings have 'strong implications for many movement disorders' (eg in the Impact Statement). What are these implications, can they be spelled out? Similarly, in the Discussion it is stated that "this has major implications for emerging theories of movement cancellation". What are these implications?

First, we note that button presses in the stop-signal task are not ballistic (de Jong et al., 1990). However, we understand and appreciate the reviewer’s point that some successful stops may happen “earlier” in the brain or in the corticospinal motor system, before innervation reaches the peripheral muscle. Accordingly, we have changed the phrasing from “already-initiated” to “prepotent” on p. 4.

Second, because there is motor inhibition at baseline and the dominant models of motor control state that this baseline inhibition is reduced to enable movement, we consider the relative increase in beta burst rates after stop-signals to be a “re-instantiation” of the inhibition that is present at baseline.

Third, we agree that the implications for understanding of movement disorders and their treatment should be better spelled out. We have added to that paragraph on p. 29 of the Discussion to expand upon those implications.

14) In the explanation at the bottom of page 13, it is argued that the participant-specific SSRT accounts for the differences between analyses. Could the difference also simply be explained by one analysis involving (arbitrary) binning, while the other integrates differences across several bins into a single time window?

Indeed, we believe that artificially binning the data can account for the absence of findings in that analysis in several ways. One is the latency differences in processing indicated by differences in SSRT. Another are the (biologically meaningless) boundaries of the bins. Finally, it could be a difference in power of the binning-based analysis. For all of those reasons, we consider the analysis that considers the entire SSD-SSRT window (Figure 3B) to be the superior approach, as we note in the text.